# A flexible $z-$layers approach for the accurate representation of free surface flows in a coastal ocean model (SHYFEM v. 7_5_71)

Luca Arpaia[1], Christian Ferrarin[1], Marco Bajo[1], and Georg Umgiesser[1,2]

[1]Institute of Marine Sciences, National Research Council, Castello 2737/F, 30122 Venice, Italy
[2]Klaipėda University, Marine Research Institute, H.Manto 84, 92294 Klaipėda, Lituania

**Correspondence:** Luca Arpaia (luca.arpaia@ve.ismar.cnr.it)

**Abstract.** We propose a discrete multilayer shallow water model based on $z-$layers which, thanks to the insertion and removal of surface layers, can deal with an arbitrarily large tidal oscillation independently of the vertical resolution. The algorithm is based on a classical two-step procedure used in numerical simulations with moving boundaries (grid movement followed by a grid topology change, that is the insertion/removal of surface layers) which avoids the appearance of surface layers with very small or negative thickness. With ad-hoc treatment of advection terms at non-conformal edges that may appear due to insertion/removal operations, mass conservation and the compatibility of the tracer equation with the continuity equation are preserved at a discrete level. This algorithm called $z-$surface-adaptive, can be reduced, as a particular case when all layers are moving, to the $z-$star coordinate. With idealized and realistic numerical experiments, we compare the $z-$surface-adaptive against $z-$star and we show that it can be used to simulate effectively coastal flows.

## 1 Introduction

The accuracy of ocean models in reproducing many dynamical processes is highly related to their vertical coordinate system. In literature, many choices exist covering the spectrum of coordinate systems. There are four main types of vertical coordinates which correspond to different vertical subdivisions of the fluid domain: 1) isopycnal layers with the interfaces that are material surfaces (Lagrangian framework); 2) $z-$layers with fixed interfaces parallel to geopotentials (Eulerian framework); 3) terrain/surface-following $\sigma$ or $s$-layers with interfaces adapted to the ocean surface and bottom boundaries; 4) adaptive coordinate with interfaces that dynamically adapt to better capture different flow features (Lagrangian tendencies, stratification and shear). The last two types move "arbitrarily" with respect to the flow, either to adapt to the free surface or any other features, and belong to the Arbitrary Lagrangian Eulerian framework (ALE).

$z-$layers were used in early ocean and coastal models and are nowadays implemented and used in some ocean models (HAMSOM, Backhaus, 1985), (TRIM-3D, Cheng et al., 1993), (UNTRIM-3D, Casulli and Walters, 2000), (SHYFEM, Umgiesser, 2022). They are attractive when simulating strongly stratified flows (Hordoir et al., 2015) and low frequency motions (Leclair and Madec, 2011). This occurs because the isopycnals are well aligned to the $z-$interfaces or they slowly depart from them. At the same time, the truncation error of the internal pressure gradient term remains very small.

A vertical discretization based on fixed interfaces is expected to have issues with the complex and moving boundaries represented by the free surface and by the ocean bottom. In this manuscript, we focus on $z-$layers performances relative to the treatment of the free surface boundary. To simplify the boundary condition at the free surface, $z-$layers were typically coded allowing the surface layer to vary in thickness (Griffies et al., 2001). However, in such models, the surface layer cannot vanish, which implies that the free surface variation must be smaller than the surface layer thickness. For coastal applications, this is a serious drawback, especially for the vertical resolution in shallow areas with high tidal elevations. In order to overcome this problem, other $z-$type coordinates have been introduced over the years: they are based on $z-$layers that move to accommodate the tidal oscillation, but the bottom is not a coordinate surface (they are surface-following but not terrain-following). These coordinates are clearly of ALE-type but in the ocean modelling literature they are classified as $z$ because the deviation from the geopotentials is very small. They combine small diapycnal mixing, especially for internal tides computations, and small truncation error on the pressure gradient term. The $z-$star of Adcroft and Campin (2004), the quasi$-z$ of Mellor et al. (2002) and the hybrid $z/\sigma$ of Burchard and Petersen (1997) all belong to such $z-$*surface-following* system. An alternative to deal with the moving surface is to keep the vertical grid perfectly aligned to geopotentials, thus working in a truly Eulerian framework, but allowing the surface layer(s) to be removed or inserted. We refer to this system as $z-$*surface-adaptive*. Insertion/removal of the surface layer has been discussed in Casulli and Cheng (1992) and it is used for example in Burchard and Baumert (1998). However "both the accuracy and stability are suspect; it is most likely difficult to make the transition of a vanishing layer smooth enough to not generate numerical problems; conservation issues are a major concern and the likelihood of vanishing layers become more frequent with increasing vertical resolution" (Adcroft and Campin, 2004).

In this paper we propose an algorithm for the $z-$surface adaptive coordinate which goes beyond such limitations. We employ a classical grid adaptation strategy for situations in which the adaptation is driven by a moving boundary (Guardone et al., 2011). It combines a first ALE grid movement step (surface interface displacement stretched by the free surface displacement) and a second topology modification step (layer insertion, layer removal). All these operations are easily performed on the one-dimensional vertical grid. If the water depth is positive, the thickness of the surface layers remains positive, avoiding stability issues related to the appearance of small or negative layers. We show that the mass is conserved. Also the discrete preservation of a constant tracer can be easily accomplished, which guarantees a complete consistency at a discrete level of the tracer equation with the continuity equation as shown since the work of Lin and Rood (1996); Gross et al. (2002).

This solution generalizes $z-$layers in the sense that the same algorithm can be easily reduced to $z-$star and can be added to a flexible vertical coordinate system. In fact, the grid adaptation has one free parameter that controls the number of moving layers. Tuning such a parameter, so that all the layers along the water column are moving, we show the link of the proposed approach with the $z-$star.

The algorithm is implemented in the SHYFEM finite-element ocean model of the CNR-ISMAR (Umgiesser et al. (2004), https://github.com/SHYFEM-model/shyfem) which implements the multilayer shallow water equations with $z$ and $\sigma$ layers. SHYFEM uses a semi-implicit finite element discretization on unstructured B-type grids derived from the work of Casulli and Cheng (1992) and Williams and Zienkiewicz (1981).

The manuscript is organized as follows: in Section 2 we introduce the vertical discretization and the multilayer shallow water model. Three different vertical discretizations are considered: the standard multilayer shallow water model based on $\sigma$-layers, then the $z-$star and the $z$-layers. In Section 3 we provide the semi-implicit finite element discretization of the multilayer equations. In Section 4 we describe the $z-$surface-adaptive algorithm, in Section 5 we detail the issue of a spatially variable number of surface layers caused by the insertion/removal operations. In Section 6 we provide numerical tests and in Section 7 we conclude with a discussion.

## 2    Multilayer shallow water model

We start considering the multilayer (or layer integrated) shallow water model for stratified flows studied in Audusse et al. (2011). The space variable is $(\boldsymbol{x}, z) \in \mathbb{R}^3$ with $\boldsymbol{x} = (x, y) \in \mathbb{R}^2$ that denotes the horizontal space variable. We consider the fluid domain $\Omega$:

$$\Omega = \Big\{ (\boldsymbol{x}, z) : \boldsymbol{x} \in \Omega_{\boldsymbol{x}}, \; -z_b(\boldsymbol{x}) \leq z \leq \zeta(\boldsymbol{x}, t) \Big\}$$

where $\Omega_{\boldsymbol{x}}$ is the projection of $\Omega$ onto the horizontal plane, $\zeta(\boldsymbol{x}, t)$ is a function that represents the free surface elevation and $z_b(\boldsymbol{x})$ is the bathymetry that does not depend on time. The water depth is $H(\boldsymbol{x}, t) = \zeta(\boldsymbol{x}, t) + z_b(\boldsymbol{x})$. As depicted in Figure 1, right panel, the multilayer shallow water model is based on a discretization of the domain $\Omega$ with a vertical grid composed of $N$ layers denoted $\Omega_\alpha$ with $\alpha = 1, ..., N$, ordered from the free surface to the bottom. The layers are non-overlapping with $\Omega = \bigcup_{\alpha=1}^N \Omega_\alpha$. Each layer $\Omega_\alpha$ is delimited laterally by the vertical domain boundary and in the vertical by the time dependent interfaces $\Gamma_{\alpha \pm 1/2}(t)$ defined by the set of points of coordinates $(\boldsymbol{x}, z)$ such that $z = z_{\alpha \pm 1/2}(\boldsymbol{x}, t)$. The free surface $\Gamma^\zeta$ and the bottom interfaces $\Gamma^b$ are described respectively by the free surface elevation $z_{1/2} = \zeta(\boldsymbol{x}, t)$ and by the bathymetry function $z_{N+1/2} = -z_b(\boldsymbol{x})$. In order to provide the rules for such slicing of the domain, we define a reference domain which is constant in time, with space variables $(\boldsymbol{x}, s) \in \mathbb{R}^3$ such that:

$$\Omega^0 = \Big\{ (\boldsymbol{x}, s) : \boldsymbol{x} \in \Omega_{\boldsymbol{x}}, \; -1 \leq s \leq 0 \Big\}$$

and discretized by means of a vertical grid similarly composed of $N$ layers, each denoted $\Omega_\alpha^0$. The reference layers are delimited vertically by the fixed-in-time interfaces $\Gamma_{\alpha \pm 1/2}^0$, which are placed at the vertical coordinate given by the coefficients $s_{\alpha \pm 1/2}$. Such constants can be ordered:

$$s_{1/2} = 0 < s_{2-1/2} < ... < s_{N+1/2} = -1$$

Then the interface position can be obtained by mapping the reference interface $\Gamma_{\alpha-1/2}^0$ to the actual or physical interface $\Gamma_{\alpha-1/2}(t)$. In general we assume that exists a function, for $\alpha = 1, ..., N$:

$$A : \Gamma_{\alpha-1/2}^0 \to \Gamma_{\alpha-1/2}(t), \quad z_{\alpha-1/2} = A(\boldsymbol{x}, s_{\alpha-1/2}, t) \quad \boldsymbol{x} \in \Omega_{\boldsymbol{x}} \tag{1}$$

To prescribe this function we use the generalized vertical coordinate transformation, see Mellor et al. (2002):

$$z_{\alpha-1/2} = \zeta(\boldsymbol{x}, t) + s_{\alpha-1/2} \left( \zeta(\boldsymbol{x}, t) + z_b(\boldsymbol{x}) \right) \tag{2}$$

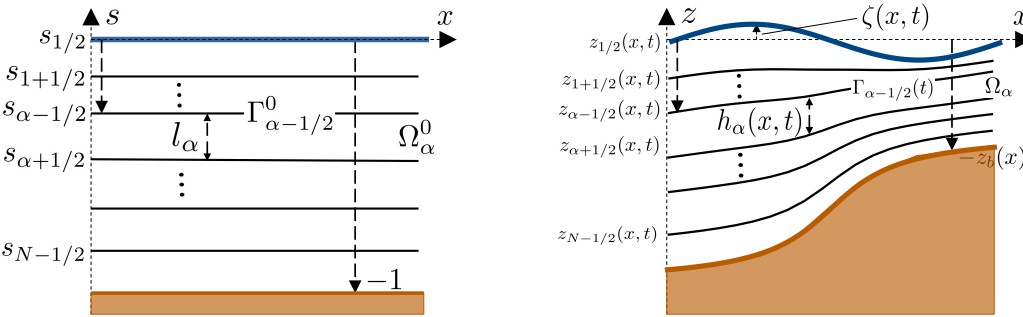

**Figure 1.** One-dimensional sketch of the reference (left) and physical (right) domains for the multilayer shallow water model.

which assures a surface and terrain-following grid that is limited by the interfaces $\Gamma_{1/2}(t) = \Gamma^\zeta(t)$ and $\Gamma_{N+1/2} = \Gamma^b$. The reference and the physical domains with their vertical subdivisions are sketched in Figure 1. Using this transformation, the layer thickness can be deduced from the water depth, for $\alpha = 1, ..., N$:

$$h_\alpha(\boldsymbol{x},t) \;=\; z_{\alpha-1/2}(\boldsymbol{x},t) - z_{\alpha+1/2}(\boldsymbol{x},t) \tag{3}$$

$$\;=\; \left(s_{\alpha-1/2} - s_{\alpha+1/2}\right) H(\boldsymbol{x},t) = l_\alpha H(\boldsymbol{x},t) \tag{4}$$

where the coefficients $l_\alpha = s_{\alpha-1/2} - s_{\alpha+1/2}$ are prescribed after the creation of the reference grid. They are positive and they sum to one $\sum_{\alpha=1}^{N} l_\alpha = 1$. The multilayer model is based on a piecewise constant approximation, on the vertical grid, of the horizontal fluid velocity and of a generic tracer. For $\alpha = 1, ..., N$:

$$\boldsymbol{u}_\alpha(\boldsymbol{x},t) \;=\; \frac{1}{h_\alpha} \int\limits_{z_{\alpha+1/2}}^{z_{\alpha-1/2}} \boldsymbol{u}(\boldsymbol{x},z,t)\, dz \tag{5}$$

$$T_\alpha(\boldsymbol{x},t) \;=\; \frac{1}{h_\alpha} \int\limits_{z_{\alpha+1/2}}^{z_{\alpha-1/2}} T(\boldsymbol{x},z,t)\, dz \tag{6}$$

The tracer for us will be the salinity. We assume that the fluid density depends on salinity through the Unesco equation of state (Millero and Poisson, 1981) at one atmosphere and at a constant potential temperature of $12.4\,^\circ C$. If the equation of state is of type $\rho = \rho(T)$, the density vertical discretization derives from the tracer one, for $\alpha = 1, ..., N$:

$$\rho_\alpha(\boldsymbol{x},t) \;=\; \rho(T_\alpha(\boldsymbol{x},t)) \tag{7}$$

We introduce the following notation for a generic function $f(z)$:

- To express a function that is discontinuous at the interface, we use the same notation of Fernández-Nieto et al. (2014):

$$f_{\alpha-1/2}^{+} \;=\; \left(f|_{\Omega_\alpha}\right)_{\Gamma_{\alpha-1/2}}, \quad f_{\alpha-1/2}^{-} = \left(f|_{\Omega_{\alpha-1}}\right)_{\Gamma_{\alpha-1/2}}$$

– if the function is continuous

$$f_{\alpha-1/2} = f^+_{\alpha-1/2} = f^-_{\alpha-1/2} = f|_{\Gamma_{\alpha-1/2}}$$

– the difference of the function between the upper and lower interface is

$$\left[f\right]^{\alpha-1/2}_{\alpha+1/2} = f_{\alpha-1/2} - f_{\alpha+1/2}$$

Mass conservation reads:

$$\frac{\partial \zeta}{\partial t} + \nabla \cdot \left(\sum_{\beta=1}^{N} h_\beta \boldsymbol{u}_\beta\right) = 0 \tag{8}$$

In this work we consider the multilayer shallow water model for stratified fluid with the Boussinesq assumption. Momentum and tracer equations in the multilayer approach can be written for $\alpha = 1, ..., N$:

$$\frac{\partial h_\alpha \boldsymbol{u}_\alpha}{\partial t} + \nabla \cdot (h_\alpha \boldsymbol{u}_\alpha \otimes \boldsymbol{u}_\alpha) = \left[\boldsymbol{u}G\right]^{\alpha-1/2}_{\alpha+1/2} - gh_\alpha \nabla\zeta + \left[\boldsymbol{K}\right]^{\alpha-1/2}_{\alpha+1/2} + \boldsymbol{B}_\alpha \tag{9}$$

$$\frac{\partial h_\alpha T_\alpha}{\partial t} + \nabla \cdot (h_\alpha T_\alpha \boldsymbol{u}_\alpha) = \left[TG\right]^{\alpha-1/2}_{\alpha+1/2} + \left[K_T\right]^{\alpha-1/2}_{\alpha+1/2} \tag{10}$$

where $G_{\alpha\pm1/2}$ is the mass-transfer function responsible for the vertical mass exchange between the layers, $\boldsymbol{K}_{\alpha\pm1/2}$ are the vertical viscous fluxes that model the shear stress between the layers, $K_{T,\alpha\pm1/2}$ are the vertical diffusive fluxes that model the diffusive process between the layers, $\boldsymbol{B}_\alpha$ models the pressure force related to the buoyancy gradient. The system (8)(9) and (10) is implemented in the SHYFEM model, as well as in many other ocean models (Burchard and Petersen, 1997; Klingbeil et al., 2018). If $N$ is the number of vertical layers, the equations are solved for $2N+1$ unknown variables, which are: the free surface elevation, the layer discharges $h_\alpha \boldsymbol{u}_\alpha$ and the layer-integrated tracer $h_\alpha T_\alpha$. The layer thickness is deduced from the water depth through equation (4). In the following, we give the details of the SHYFEM implementation of each term on the right-hand side.

From the derivation of Fernández-Nieto et al. (2014), the definition of the mass-transfer function is:

$$
\begin{aligned}
G_{\alpha-1/2} &= \left(\nabla z_{\alpha-1/2} \cdot \boldsymbol{u}_\alpha\right) + \sigma_{\alpha-1/2} - w^+_{\alpha-1/2} \\
&= \left(\nabla z_{\alpha-1/2} \cdot \boldsymbol{u}_{\alpha-1}\right) + \sigma_{\alpha-1/2} - w^-_{\alpha-1/2}
\end{aligned}
\tag{11}
$$

with $\sigma_{\alpha-1/2}$ the velocity of the grid interface:

$$\sigma_{\alpha-1/2} = \frac{\partial z_{\alpha-1/2}}{\partial t} \tag{12}$$

and $w^\pm_{\alpha-1/2}$ the vertical fluid velocity at the interface. The vertical velocity is computed from the following relationships:

$$w^+_{\alpha-1/2} = -w^-_{\alpha+1/2} - h_\alpha \nabla \cdot \boldsymbol{u}_\alpha \quad \text{and} \quad w^-_{\alpha-1/2} = w^+_{\alpha-1/2} + \nabla z_{\alpha-1/2} \cdot (\boldsymbol{u}_\alpha - \boldsymbol{u}_{\alpha-1}) \tag{13}$$

which are evaluated starting from the bottom $\alpha = N, ..., 1$, where the no slip condition is imposed $w^-_{N+1/2} = \boldsymbol{u}_N \cdot \nabla z_b$. In practice and as it is standard in ocean models, the mass-transfer function is computed directly from the layer-integrated mass equation

$$G_{\alpha-1/2} = G_{\alpha+1/2} + \frac{\partial h_\alpha}{\partial t} + \nabla \cdot (h_\alpha \boldsymbol{u}_\alpha) \tag{14}$$

Summing from $N$ to $\alpha$ as:

$$G_{\alpha-1/2} = G_{N+1/2} + \sum_{\beta=N}^{\alpha} \frac{\partial h_\beta}{\partial t} + \sum_{\beta=N}^{\alpha} \nabla \cdot (h_\beta \boldsymbol{u}_\beta) \tag{15}$$

which implies $G_{1/2} = 0$ or no mass loss at the free surface. The vertical velocity at the interfaces $w^\pm_{\alpha-1/2}$ no more appears in the system but it can be computed from the incompressibility condition (13) in a post-processing step. With a horizontal velocity and tracer discontinuous at the interfaces, the vertical momentum flux in (9) is computed with a numerical flux. An upwind flux is used in this study, for $\Gamma_{\alpha-1/2}$ it reads:

$$G_{\alpha-1/2}\boldsymbol{u}_{\alpha-1/2} = G^+_{\alpha-1/2}\boldsymbol{u}_\alpha + G^-_{\alpha-1/2}\boldsymbol{u}_{\alpha-1}$$

with $G^+_{\alpha-1/2} = \max(0, G_{\alpha-1/2})$ and $G^-_{\alpha-1/2} = \min(0, G_{\alpha-1/2})$. For the tracer, a TVD flux is employed (LeVeque, 2002).

The terms $\boldsymbol{K}_{\alpha-1/2}$ and $K_{T,\alpha-1/2}$ are the vertical viscous and diffusive fluxes computed at the interface $\Gamma_{\alpha-1/2}$:

$$\boldsymbol{K}_{\alpha-1/2} = \nu_{\alpha-1/2} D_z \boldsymbol{u}_{\alpha-1/2}$$

$$K_{T,\alpha-1/2} = \nu_{T,\alpha-1/2} D_z T_{\alpha-1/2}$$

where $\nu_{\alpha-1/2}$ is the vertical viscosity and $\nu_{T,\alpha-1/2}$ the vertical diffusivity. $D_z(\cdot)$ is an approximation of the vertical derivative evaluated at the interface and resolved with finite differences. The vertical viscosity and diffusivity can be laminar or computed with a turbulence model. The bottom momentum flux is specified by a quadratic friction model. Then, the viscous fluxes read:

$$\boldsymbol{K}_{\alpha-1/2} = \begin{cases} 0, & \alpha = 1 \\ \nu_{\alpha-1/2} \frac{\boldsymbol{u}_{\alpha-1} - \boldsymbol{u}_\alpha}{(h_{\alpha-1}+h_\alpha)/2}, & \alpha = 2, ..., N \\ \boldsymbol{\tau}_b = -C_F |\boldsymbol{u}_N| \boldsymbol{u}_N, & \alpha = N+1 \end{cases}$$

with $C_F$ the bottom friction coefficient. Similarly, the diffusive fluxes read:

$$K_{T,\alpha-1/2} = \begin{cases} 0, & \alpha = 1 \\ \nu_{T,\alpha-1/2} \frac{T_{\alpha-1} - T_\alpha}{(h_{\alpha-1}+h_\alpha)/2}, & \alpha = 2, ..., N \\ 0, & \alpha = N+1 \end{cases}$$

with no tracer fluxes through the free surface and the bottom.

Finally, the term $\boldsymbol{B}_\alpha$ represents the internal pressure gradient force. The layer-integrated pressure gradient term $\int_{z_{\alpha+1/2}}^{z_{\alpha-1/2}} \nabla p(z) \, dz$, instead of applying the Leibniz rule (Audusse et al., 2011), has been split into the external pressure gradient, related to the free

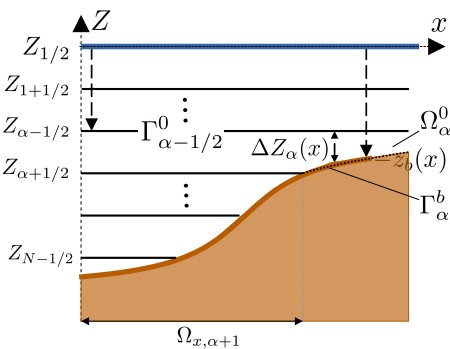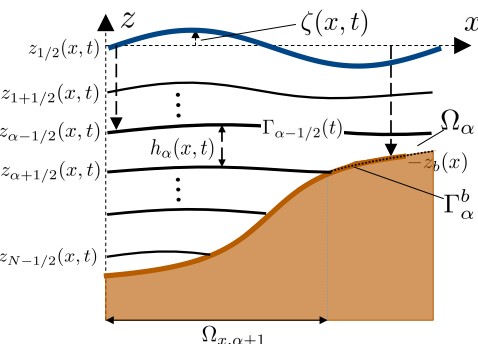

**Figure 2.** Figure. One-dimensional sketch of the reference (left) and physical (right) domains for the multilayer shallow water model with $z$-star layers.

surface slope, and the internal pressure gradient, related to the buoyancy gradient. The internal pressure gradient term is written
in the density Jacobian form of Song (1998):

$$\boldsymbol{B}_\alpha = h_\alpha b_1 \nabla \zeta + h_\alpha \sum_{\beta=1}^{\alpha} \boldsymbol{J}(b_{\beta-1/2}, z_{\beta-1/2}) h_{\beta-1/2}$$

where $h_{\beta-1/2}$ is the distance between the layer centers, that is $h_{\beta-1/2} = (h_{\beta-1} + h_\beta)/2$ for $\beta = 2, ..., N$ and $h_{\beta-1/2} = h_1/2$ for $\beta = 1$. The summation over the layers corresponds to vertical integration of the density Jacobian based on the piecewise constant profile of the density with the quadrature points placed at the interfaces. The density Jacobian at the interface is:

$\quad \boldsymbol{J}(b_{\beta-1/2}, z_{\beta-1/2}) = \nabla b_{\beta-1/2} - D_z(b_{\beta-1/2}) \nabla z_{\beta-1/2}$

If $b_\beta = g \frac{\rho_0 - \rho_\beta}{\rho_0}$ is the layer buoyancy, the buoyancy at the interface is resolved with an average $b_{\beta-1/2} = \frac{1}{2}(b_{\beta-1} + b_\beta)$ for $\beta = 2, ..N$ and $b_{\beta-1/2} = b_1$ for $\beta = 1$. The approximation of the vertical derivative evaluated at the interface is resolved with finite differences. It is taken zero for the first interface $D_z(b_{\beta-1/2}) = 0$ for $\beta = 1$ and $D_z(b_{\beta-1/2}) = (b_{\beta-1} - b_\beta)/h_{\beta-1/2}$ for $\beta = 2, ..., N$. These choices allow us to recover a standard formula that can be found in Shchepetkin and McWilliams (2003)
or in Klingbeil et al. (2018).

The tracer equation (10) admits a trivial solution which we want to inherit also at the discrete level, the so-called tracer constancy condition: for a constant tracer, equation (10) reduces to the layerwise mass equation (14). The importance of preserving this property at a discrete level has been discussed extensively in Gross et al. (2002).

## 2.1 $z-$star

The multilayer model presented so far is based on a vertical subdivision of the fluid domain through the surface/ terrain-following transformation (2) which leads to the coefficients $l_\alpha$ given in (4). Other vertical subdivisions can be used leading to different coefficients that, however, must verify both the positivity constraint and they have to sum to one. In the following, we specify a slicing of the domain with both these properties based on a vertical coordinate transformation called $z-$star (Adcroft

and Campin, 2004). The reference domain, with vertical coordinate $Z$, is:

$$\Omega^0 = \left\{ (\boldsymbol{x}, Z) : \boldsymbol{x} \in \Omega_{\boldsymbol{x}}, \ -z_b(\boldsymbol{x}) \leq Z \leq 0 \right\}$$

This domain is discretized by means of a vertical grid composed of $N$ layers, with interfaces $\Gamma^0_{\alpha-1/2}$, which are aligned to the geopotential. These interfaces can be described by constant functions:

$$Z_{1/2} = 0 < Z_{2-1/2} < ... < Z_{N+1/2} = -\max z_b(\boldsymbol{x})$$

As shown in Figure 2, there is a substantial difference with respect to the vertical subdivision of the terrain-following grid. The
180 grid interfaces could intersect the bathymetry and should be defined only in the fluid domain. We define the projection of the interface $\Gamma^0_{\alpha-1/2}$ onto the horizontal plane as:

$$\Omega_{\boldsymbol{x},\alpha} = \left\{ \boldsymbol{x} : \boldsymbol{x} \in \Omega_{\boldsymbol{x}} \text{ and } -z_b(\boldsymbol{x}) \leq Z_{\alpha-1/2} \right\} \tag{16}$$

If a layer is bounded laterally by the bathymetry interface we can denote this lateral land boundary of the layer as :

$$\Gamma^b_\alpha \quad = \quad \left\{ (\boldsymbol{x}, Z) : \ Z = -z_b(\boldsymbol{x}) \text{ and } Z_{\alpha+1/2} \leq Z \leq Z_{\alpha-1/2} \right\}$$

Each layer $\Omega^0_\alpha$ results delimited on the upper and bottom side by $\Gamma^0_{\alpha \mp 1/2}$ and laterally by the vertical domain boundary as well as it could be delimited by $\Gamma^b_\alpha$ (see Figure 2, right panel). To map the reference interface $\Gamma^0_{\alpha-1/2}$ to the physical interface $\Gamma_{\alpha-1/2}$, again, we can use a generalized coordinate transformation, for $\alpha = 1, ..., N$:

$$z_{\alpha-1/2} = \zeta(\boldsymbol{x}, t) + S_{\alpha-1/2}(\boldsymbol{x}) \left( \zeta(\boldsymbol{x}, t) + z_b(\boldsymbol{x}) \right), \qquad \boldsymbol{x} \in \Omega_{\boldsymbol{x},\alpha} \tag{17}$$

with $S_{\alpha-1/2}$ a stretching function defined as:

$$S_{\alpha-1/2}(\boldsymbol{x}) = \frac{Z_{\alpha-1/2}}{z_b(\boldsymbol{x})}$$

As in the previous Section, the layer thickness can be deduced from the total water depth. After some calculations we get:

$$\begin{aligned} h_\alpha(\boldsymbol{x}, t) \quad &= \quad z_{\alpha-1/2}(\boldsymbol{x}, t) - \max \left( z_{\alpha+1/2}(\boldsymbol{x}, t), -z_b(\boldsymbol{x}) \right) \\ &= \quad \left( S_{\alpha-1/2}(\boldsymbol{x}) - \max \left( S_{\alpha+1/2}(\boldsymbol{x}), -1 \right) \right) H(\boldsymbol{x}, t) = l_\alpha(\boldsymbol{x}) H(\boldsymbol{x}, t), \qquad \boldsymbol{x} \in \Omega_{\boldsymbol{x},\alpha} \end{aligned} \tag{18}$$

If we define $\Delta Z_\alpha(\boldsymbol{x}) = Z_{\alpha-1/2} - \max \left( Z_{\alpha+1/2}, -z_b(\boldsymbol{x}) \right)$ we can rewrite the coefficients, for $\alpha = 1, N$:

$$l_\alpha(\boldsymbol{x}) = \frac{\Delta Z_\alpha(\boldsymbol{x})}{z_b(\boldsymbol{x})}, \qquad \boldsymbol{x} \in \Omega_{\boldsymbol{x},\alpha}$$

which is prescribed once the reference grid is created. The coefficients satisfy both the positivity constraint and locally they sum to one.

An important property of the $z-$star transformation is that the horizontal domain $\Omega_{\boldsymbol{x},\alpha}$ where the layer thickness $h_\alpha$ is defined, does not depend on time, as one can verify after computing the transformation (17) for $Z_{\alpha-1/2} = -z_b(\boldsymbol{x})$. This is

particularly helpful because the number of layers does not depend on time, and the coefficients too. Other $z-$layers formulations based on similar mappings, such as the quasi$-z$ layers (Mellor et al., 2002) or the hybrid $z/\sigma$ layers (Burchard and Petersen, 1997) do not share this property. For these coordinates a special treatment of the bottom is necessary: either an *ad hoc* modification of the bottom geometry or more interestingly these coordinates could be coupled with the porosity approach recently proposed by Debreu et al. (2020) where all the layers are present in the computation. For $z-$star the bottom momentum and tracer fluxes must be properly modified, replacing the maximum number of layers $N$, with the local number of layers $N_b(\boldsymbol{x}) = \{\alpha : Z_{\alpha+1/2} < -z_b(\boldsymbol{x}) \le Z_{\alpha-1/2}\}$.

## 2.2  $z-$layers

For $z-$layers the actual interfaces do not depend on time and space:

$$z_{\alpha-1/2} = Z_{\alpha-1/2}$$

This method is implemented in the ocean models by allowing the top layer to vary in thickness without vanishing (Griffies et al., 2001). For the above transformation with fixed interfaces, the mass-transfer function (eq. (14)) coincides with the vertical velocity:

$$G_{\alpha-1/2} = -w^-_{\alpha-1/2} = -w^+_{\alpha-1/2}, \quad \alpha = 2, N+1$$

Replacing the mass transfer function with the vertical velocity in the multilayer model, we obtain the Eulerian model of Rambaud (2011).

## 3  Semi-implicit staggered finite element discretization

The discretization for both the $z-$star and the $z-$layers shallow water model can proceed in an equivalent fashion. We consider a discretization of the horizontal domain $\Omega_{\boldsymbol{x}} \in \mathbb{R}^2$ composed by non-overlapping triangular elements. We denote the horizontal grid by $\mathcal{T}$ with $K \in \mathcal{T}$ the generic triangle, $|K|$ its area. The local reference element length is $\mathrm{h}_K$ and it is computed as the minimum length of the triangle sides. With $i \in \mathcal{T}$ we denote the nodes of the grid. When no confusion is generated, we will locally number as ($j = 1, 2, 3$ or $j \in K$) the nodes of the generic triangle. Given a node $i$ in an element $K$, $\boldsymbol{n}_i^K$ denotes the inward vector normal to the edge of $K$ opposite to $i$, scaled by the length of the edge, see Figure 3, left panel. For every node of the triangulation, $\mathcal{D}_i$ denotes the subset of triangles containing $i$. The dual cell $C_i$ is obtained by joining the barycenters of the triangles in $\mathcal{D}_i$ with the midpoints of the edges meeting in $i$ as illustrated in Figure 3, middle panel. Its area is

$$|C_i| = \sum_{K \in \mathcal{D}_i} \frac{|K|}{3}$$

delimited by the boundary $\partial C_i$. The edge of $\partial C_i$ separating $C_i \cap K$ and $C_j \cap K$ has an exterior normal called $\boldsymbol{n}_{ij}^K$, as illustrated in Figure 3, left panel. As before it is scaled by the edge length. Moreover, due to the definition of the dual cell, we have:

$$\sum_{j \in K, j \neq i} \boldsymbol{n}_{ij}^K = -\frac{\boldsymbol{n}_i^K}{2} \tag{19}$$

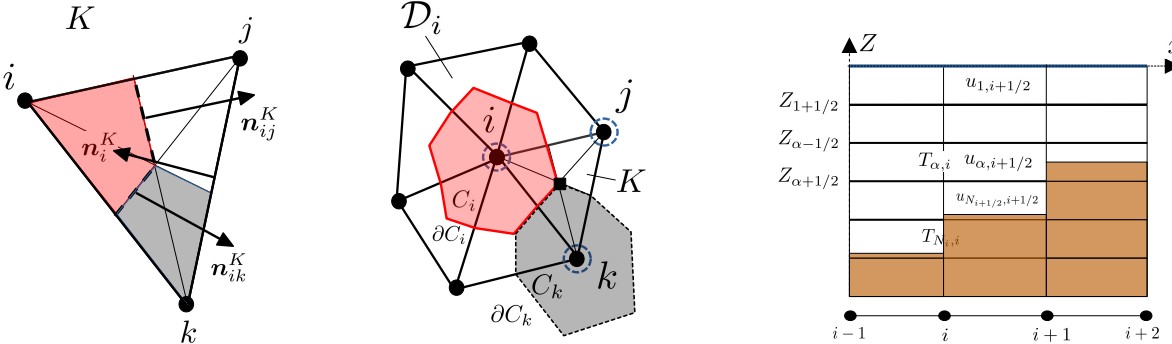

**Figure 3.** Grid and notation. Left: triangle $K$ with nodes and scaled normals. Middle: set $\mathcal{D}_i$ with dual cell area $C_i$ and dual cell boundary $\partial C_i$. The degrees of freedom are also shown: discharge ■, tracer and free surface ◯. Right: stepped bathymetry with masked boxes in brown, after the horizontal discretization.

After the horizontal discretization, the domain results subdivided into prismatic boxes $K \times [z_{\alpha+1/2}, z_{\alpha-1/2}]$. At the bottom,
$z-$layers models apply a mask to non-existing land boxes that makes the bathymetry stepped, as sketched in Figure 3, right panel. The bottom layer for each element will be denoted as $N_K$. For a staggered discretization it is helpful also to define a nodal bottom layer $N_i = \max_{K \in \mathcal{D}_i} N_K$. The projections of the interfaces onto the horizontal plane are still denoted as $\Omega_{\boldsymbol{x},\alpha}$ and defined with (16), this time evaluated with the stepwise approximation of the bathymetry. Then a layer dual cell $C_{\alpha,i}$ can be defined by considering $\mathcal{D}_{\alpha,i}$ the subset of elements sharing node $i$ and in $\Omega_{\boldsymbol{x},\alpha}$. Its area is

$$|C_{\alpha,i}| = \sum_{K \in \mathcal{D}_{\alpha i}} \frac{|K|}{3}$$

On a B-staggered grid the free surface elevation, the discharges and the tracers are described with basis functions of different order and support (Williams and Zienkiewicz, 1981). The discharge field and the tracer field belong to a finite dimensional space with basis composed by the piecewise constant functions. For the discharges, the space has basis $\{\psi_K\}_{K \in \mathcal{T}}$ composed by the characteristic functions on the triangle, while for the tracers we choose $\{\phi_i\}_{i \in \mathcal{T}}$ composed by the characteristic functions
on the dual cell. The discharge fields $\boldsymbol{q}_\alpha = h_\alpha \boldsymbol{u}_\alpha$ and the tracers $T_\alpha$ are approximated through (we use an abuse of notation employing the same symbol of the continuous variable):

$$\boldsymbol{q}_\alpha(\boldsymbol{x}, t) = \sum_{K \in \mathcal{T}} \psi_K(\boldsymbol{x}) \boldsymbol{q}_{\alpha,K}(t) \tag{20}$$

$$T_\alpha(\boldsymbol{x}, t) = \sum_{i \in \mathcal{T}} \phi_i(\boldsymbol{x}) T_{\alpha,i}(t) \tag{21}$$

with $\boldsymbol{q}_{\alpha,K}(t)$, defined for $\alpha = 1, ..., N_K$, being the elemental discharge values and with $T_{\alpha,i}(t)$, defined for $\alpha = 1, ..., N_i$, the
nodal tracer values. The free surface belongs to a space of finite dimension with basis $\{\varphi_i\}_{i \in \mathcal{T}}$ which denotes the standard continuous piecewise linear Lagrange basis. The discrete free surface is given by:

$$\zeta(\boldsymbol{x}, t) = \sum_{i \in \mathcal{T}} \varphi_i(\boldsymbol{x}) \zeta_i(t) \tag{22}$$

where $\zeta_i(t)$ are the nodal free surface values. Note that the discrete discharges and discrete tracers are discontinuous respectively across the boundaries of the triangles and of the dual cells whereas the discrete free surface is globally continuous. On a B-grid the layer thickness is naturally computed at the grid nodes $h_{\alpha,i}$, where the free surface is available. The element values $h_{\alpha,K}$ are a conservative average of the nodal values. The element velocities are obtained from $\boldsymbol{u}_{\alpha,K} = \frac{\boldsymbol{q}_{\alpha,K}}{h_{\alpha,K}}$.

We obtain the weak formulation multiplying mass and momentum equations (8) and (9) by the test functions that belong to the same space of the solution and integrating it on the horizontal domain. The finite element discretization reduces to compute the integrals accounting for the different terms. For the mass flux term, which is integrated by parts we need to compute with a proper quadrature rule the following integral (only $x-$component shown):

$$a_{iK}^x = \int_K \frac{\partial \varphi_i}{\partial x} \, d\boldsymbol{x}$$

The boundary term has been neglected since it cancels out except at the lateral domain boundary. Similarly, for the terms that will be treated explicitly in the momentum equation namely the horizontal/vertical advection and the internal pressure gradient, we have:

$$f_{\alpha,K}^x = -\int_{\partial K} \widehat{\boldsymbol{q}_\alpha u_\alpha} \cdot \boldsymbol{n} \, ds + \int_K \left( B_\alpha^x + \left[ uG \right]_{\alpha+1/2}^{\alpha-1/2} \right) d\boldsymbol{x}$$

The horizontal advection term is resolved with a first-order upwind flux $\widehat{\boldsymbol{q}_\alpha u_\alpha}$ (Umgiesser et al., 2004). To write the scheme in matrix form, exploiting the compactness of the staggered discretization, we introduce "vertical" vectors/matrices, that pile up all the layers for a single element $K$, and we denote them with bold capital letters. For example, the layer discharges and the layer thickness are regrouped in the following vectors:

$$\boldsymbol{U}_K = \begin{pmatrix} q_{1,K}^x \\ ... \\ q_{\alpha,K}^x \\ ... \\ q_{N_K,K}^x \end{pmatrix}, \quad \boldsymbol{V}_K = \begin{pmatrix} q_{1,K}^y \\ ... \\ q_{\alpha,K}^y \\ ... \\ q_{N_K,K}^y \end{pmatrix}, \quad \boldsymbol{H}_K = \begin{pmatrix} h_{1,K} \\ ... \\ h_{\alpha,K} \\ ... \\ h_{N_K,K} \end{pmatrix}$$

and analogously the explicit terms:

$$\boldsymbol{F}_K^x = \begin{pmatrix} f_{1,K}^x \\ ... \\ f_{\alpha,K}^x \\ ... \\ f_{N_K,K}^x \end{pmatrix}, \quad \boldsymbol{F}_K^y = \begin{pmatrix} f_{1,K}^y \\ ... \\ f_{\alpha,K}^y \\ ... \\ f_{N_K,K}^y \end{pmatrix}$$

The vertical viscous term is recast in matrix form via a tridiagonal matrix $\boldsymbol{A}_K^d \in \mathbb{R}^{N_K \times N_K}$. The bottom momentum flux has to be integrated into this matrix. Note that all these vectors/matrices are restricted to non-masked boxes.

Following Casulli and Cattani (1994), we build a semi-implicit time discretization by treating semi-implicitly the mass flux and the free surface gradient in the momentum equation. The vertical viscous term can also cause a restrictive time-step and is handled here implicitly without major computation issues but allowing to relax the CFL condition. We define the variation of a quantity in a time step as $\Delta u = u^{n+1} - u^n$, then:

$$u^{n+\theta} = \theta u^{n+1} + (1-\theta)u^n = \theta \Delta u + u^n$$

After applying the previous definition to the semi-discrete equations, the semi-implicit momentum equations on an unstructured B-grid read:

$$\Delta \boldsymbol{U}_K = \Delta \boldsymbol{U}_K^* - \Delta t g \boldsymbol{A}_K^{-1} \boldsymbol{H}_K^n \sum_{j \in K} a_{jK}^x \theta \Delta \zeta_j \tag{23}$$

$$\Delta \boldsymbol{V}_K = \Delta \boldsymbol{V}_K^* - \Delta t g \boldsymbol{A}_K^{-1} \boldsymbol{H}_K^n \sum_{j \in K} a_{jK}^y \theta \Delta \zeta_j \tag{24}$$

with $\boldsymbol{A}_K = \left( \boldsymbol{I}|K| - \Delta t \boldsymbol{A}_K^d \right)$ a tridiagonal, positive definite and diagonally dominant matrix. The non-linear dependence of the external pressure gradient term from $\boldsymbol{H}_K$ has been resolved by using the old value. Also the viscous matrix has been computed with frozen values at $t^n$. In $\boldsymbol{F}_K^n$ all the quantities are computed at $t^n$, including the mass-transfer function. These choices avoid solving a non-linear system at each time step. The variation $\Delta (\cdot)^* = (\cdot)^* - (\cdot)^n$ is the solution of the following Euler step with an explicit external pressure gradient:

$$\Delta \boldsymbol{U}_K^* = \Delta t \boldsymbol{A}_K^{-1} \left( \boldsymbol{F}_K^{x,n} + \boldsymbol{A}_K^d \boldsymbol{U}_K^n - g \boldsymbol{H}_K^n \sum_{j \in K} a_{jK}^x \zeta_j^n \right) \tag{25}$$

$$\Delta \boldsymbol{V}_K^* = \Delta t \boldsymbol{A}_K^{-1} \left( \boldsymbol{F}_K^{y,n} + \boldsymbol{A}_K^d \boldsymbol{V}_K^n - g \boldsymbol{H}_K^n \sum_{j \in K} a_{jK}^y \zeta_j^n \right) \tag{26}$$

If the expressions for $\Delta \boldsymbol{U}_K$ and $\Delta \boldsymbol{V}_K$, (23) and (24), are introduced into the discrete mass equation, we obtain a linear system with only the free surface coefficients as unknowns:

$$\sum_{K \in \mathcal{D}_i} \sum_{j \in K} \left( m_{ij}^K + g \theta^2 \Delta t^2 \left( a_{iK}^x \boldsymbol{1}^T \boldsymbol{A}_K^{-1} \boldsymbol{H}_K^n a_{jK}^x + a_{iK}^y \boldsymbol{1}^T \boldsymbol{A}_K^{-1} \boldsymbol{H}_K^n a_{jK}^y \right) \right) \Delta \zeta_j =$$
$$\Delta t \sum_{K \in \mathcal{D}_i} \left( a_{iK}^x \boldsymbol{1}^T \left( \theta \Delta \boldsymbol{U}_K^* + \boldsymbol{U}_K^n \right) + a_{iK}^y \boldsymbol{1}^T \left( \theta \Delta \boldsymbol{V}_K^* + \boldsymbol{V}_K^n \right) \right) \tag{27}$$

where $m_{ij}^K = \int_K \varphi_i \varphi_j \, d\boldsymbol{x}$ is the Galerkin mass matrix based on the piecewise linear Lagrange basis functions. The Galerkin mass matrix, in SHYFEM, is lumped. The vector $\boldsymbol{1} \in \mathbb{R}^{N_K}$ has all components being one.

The hydrodynamic time step flow chart is thus the following: we first perform the Euler step (25) and (26). Then we resolve the mass equation (27) and we complete momentum update with the semi-implicit step (23) and (24). Finally we compute the layer thickness at the grid nodes. For $z-$star, we use the expression (18) at the grid nodes. For the $z$-layers, the layer thickness does not change except for the first layer.

## 3.1 Mass-transfer function

After the hydrodynamic update of the previous paragraph, the discrete mass-transfer function is computed. We employ the same continuous piecewise linear approximation used for the free surface. The nodal values are computed from a finite-element mass-lumped discretization of the layerwise mass equation (14). As for the depth-integrated mass equation, the discharge is evaluated semi-implicitly. Starting from the bottom with $G_{N_i+1/2,i}^{n+1} = 0$, for $\alpha = N_i,...,1$:

$$|C_{\alpha,i}| G_{\alpha-1/2,i}^{n+1} = |C_{\alpha+1,i}| G_{\alpha+1/2,i}^{n+1} + |C_{\alpha,i}| \frac{\Delta h_{\alpha,i}}{\Delta t} - \sum_{K \in \mathcal{D}_{\alpha i}} \left( a_{iK}^x q_{\alpha,K}^{x,n+\theta} + a_{iK}^y q_{\alpha,K}^{y,n+\theta} \right) \tag{28}$$

Note that the semi-implicit discretization ensures vertical mass-conservation. Summing up (28) for all the layers and using equation (27) with a lumped Galerkin mass-matrix to cancel the right-hand side, we get the impermeability condition at the free surface $G_{1/2,i}^{n+1} = 0$. With standard $z-$layers, the contribution related to the grid velocity is zero $\Delta h_{\alpha,i} = \Delta t[\sigma_i]_{\alpha+1/2}^{\alpha-1/2} = 0$, except for the first layer.

## 3.2 Tracers

The semi-implicit update is completed with the time-stepping of the tracer. Vertical diffusion is treated implicitly and the remaining advection terms are explicit. The spatial discretization of the explicit terms implies the computation of the following integrals which account for the horizontal and vertical advection terms:

$$f_{\alpha,i} = -\int_{\partial C_{\alpha,i}} \widehat{T_\alpha q_\alpha} \cdot n \, ds + \int_{C_{\alpha,i}} \left[TG\right]_{\alpha+1/2}^{\alpha-1/2} d\boldsymbol{x}$$

where $\widehat{T_\alpha q_\alpha}$ is an appropriate numerical tracer flux across the dual cell boundary. At the lateral boundary $\partial\Omega_{\boldsymbol{x},\alpha}$, the tracer flux is zero for land boundaries while it is determined by the boundary conditions at the domain boundary. In the discussion that follows we consider only nodes that do not lie on the domain boundary. On a triangular grid the two terms read:

$$\int_{\partial C_{\alpha,i}} \widehat{T_\alpha q_\alpha} \cdot n \, ds = \sum_{K \in \mathcal{D}_{\alpha,i}} \sum_{j \in K, j \neq i} \widehat{T_\alpha q_\alpha} \cdot n_{ij}^K = \sum_{K \in \mathcal{D}_{\alpha,i}} \sum_{j \in K, j \neq i} \widehat{H}_\alpha(T_{\alpha,i}, T_{\alpha,j}) \tag{29}$$

$$\int_{C_{\alpha,i}} \left[TG\right]_{\alpha+1/2}^{\alpha-1/2} d\boldsymbol{x} = |C_{\alpha,i}| T_{\alpha-1/2,i} G_{\alpha-1/2,i} - |C_{\alpha+1,i}| T_{\alpha+1/2,i} G_{\alpha+1/2,i} \tag{30}$$

with $\widehat{H}_\alpha(T_{\alpha,i}, T_{\alpha,j})$ being the numerical flux in the horizontal direction and $T_{\alpha+1/2,i} G_{\alpha+1/2,i}$ the numerical flux in the vertical direction. The SHYFEM model implements second-order consistent TVD fluxes in both directions.

Using the notation with bold capital letters denoting "vertical" vectors, the tracer values and the explicit term at the nodes are regrouped in the following:

$$\boldsymbol{T}_i = \begin{pmatrix} T_{1,i} \\ \cdots \\ T_{\alpha,i} \\ \cdots \\ T_{N_i,i} \end{pmatrix}, \quad \boldsymbol{F}_i = \begin{pmatrix} f_{1,i} \\ \cdots \\ f_{\alpha,i} \\ \cdots \\ f_{N_i,i,} \end{pmatrix}$$

Vertical diffusion can also be assembled in matrix form through the discrete matrix $\boldsymbol{A}_i^d \in \mathbb{R}^{N_i \times N_i}$. Then, the discretization of the layerwise tracer equation (10) read:

$$\boldsymbol{A}_i \boldsymbol{T}_i^{n+1} = \mathbf{Diag}\{|C_{\alpha,i}| h_{\alpha,i}^n\} \boldsymbol{T}_i^n + \Delta t \boldsymbol{F}_i^n \tag{31}$$

with $\boldsymbol{A}_i = \left(\mathbf{Diag}\{|C_{\alpha,i}| h_{\alpha,i}^{n+1}\} - \Delta t \boldsymbol{A}_i^d\right)$ the vertical tracer matrix. Although the advection terms are explicit, it should be noted that the horizontal numerical fluxes are computed with the discharges evaluated at $\boldsymbol{q}_\alpha^{n+\theta}$ while the vertical numerical flux uses the last available mass-transfer function $G_{\alpha\pm1/2}^{n+1}$ from (28). This choice is important to maintain the consistency of the discrete tracer equation with the layerwise mass equation. In fact inserting a constant tracer in equation (31), yields exactly the discrete layerwise mass equation (28). The proof is left in the Appendix.

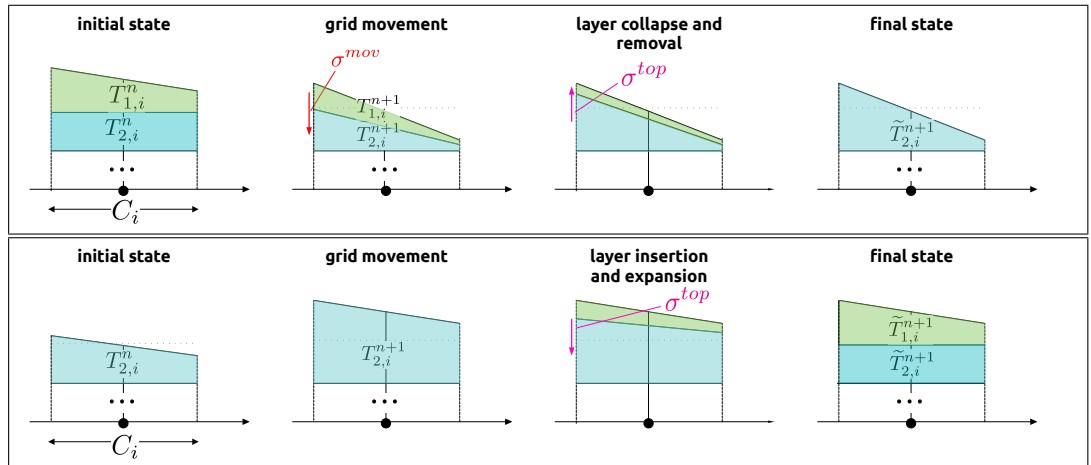

**Figure 4.** Grid and tracer evolution during one time step. The process is interpreted as four stages which bring from the pair $(T^n, \zeta_h^n)$ to $(\widetilde{T}^{n+1}, \zeta^{n+1})$. The vector $T = \{T_1, T_2\}$ collects the layer values of the tracer. Dashed line means removed interface. Left: case of surface layer insertion. Right: case of surface layer removal.

To conclude, we summarize the whole time step flow chart: after the hydrodynamic update described in Section 3, we compute the mass-transfer function (28) and, lastly, we update the tracers with (31).

## 4  $z-$surface-adaptive layers

In this section, we enhance the $z-$layers shallow water model by introducing a new algorithm that allows for the dynamic insertion and removal of surface boxes or, with an abuse of language, of surface layers. To differentiate it from the standard $z-$layers, we will refer to this enhanced version as $z-$surface-adaptive layers. The key idea is to interpret the area swept by the layer interface in the time step $\Delta t \in [t^n, t^{n+1}]$ as the sum of two contributions: one due to the mesh movement driven by the free surface oscillation (grid movement) and one due to the collapse/expansion of the layer (topology change). These topology changes in fact can be seen as a continuous deformation of the layer interfaces performed within the time step. With this in mind, the final position of the interfaces at the grid nodes $\widetilde{z}_{\alpha-1/2,i}^{n+1} = \widetilde{z}_{\alpha-1/2}(x_i, t^{n+1})$ is:

$$\widetilde{z}_{\alpha-1/2,i}^{n+1} = z_{\alpha-1/2,i}^{n+1} + \Delta\widetilde{z}_{\alpha-1/2,i}$$

where $z_{\alpha-1/2,i}^{n+1} = z_{\alpha-1/2}(x_i, t^{n+1})$ is the interface position after the grid movement and $\Delta\widetilde{z}_{\alpha-1/2,i}$ is the contribution of the interface collapse/expansion, basically a correction term. Similarly, the grid velocity in the time step can be decomposed as:

$$\sigma_{\alpha-1/2,i} = \frac{\widetilde{z}_{\alpha-1/2,i}^{n+1} - z_{\alpha-1/2,i}^n}{\Delta t} = \sigma_{\alpha-1/2,i}^{mov} + \sigma_{\alpha-1/2,i}^{top}$$

with:

$$\sigma_{\alpha-1/2,i}^{mov} = \frac{z_{\alpha-1/2,i}^{n+1} - z_{\alpha-1/2,i}^n}{\Delta t}, \quad \sigma_{\alpha-1/2,i}^{top} = \frac{\Delta\widetilde{z}_{\alpha-1/2,i}}{\Delta t}$$

In the solution of the multilayer shallow water equations we employ a splitting procedure, where the two aforementioned contributions are treated in two steps. In the first step (grid movement) we solve the multilayer model on a vertical grid where the surface layers adjust locally in order to maintain a positive thickness. In the subsequent step, we locally remove surface fluid boxes with minimal thickness or split fluid boxes that are excessively thick. The evolution of the vertical grid and of the tracer in one time step is shown in Figure 4. The top row shows the case of a layer removal and the bottom row the case of a layer

insertion. As a remark, we stress that the above interpretation of the interface displacement reveals many beneficial aspects with respect to the direct insertion and removal of a layer. Without the grid movement step, it would be more complicated to time step the tracers on a grid with positive layer thickness, with all the related stability issues. In fact in the tracer update (24) the layer thickness at $t^{n+1}$ is needed. One may think to compute the tracer after the insertion/removal operations have been performed (thus having positive layer thickness both at $t^n$ and $t^{n+1}$), but then the configuration on which the discrete tracer

equation is solved is ambiguous and it seems hard to ensure the consistency with the continuity or to verify the tracer constancy property.

In the following we provide the technical details to realize such adaptation to the free surface with the $z$-layers. First we notice that, since the beginning of the simulation, the index of the surface layer may change spatially at the element boundaries. Given the initial free surface elevation $\zeta^0(\boldsymbol{x})$, we define a set of active indices and the surface layer index, by element, as:

$$360 \quad \boldsymbol{\alpha}_{active,K} = \left\{ \alpha \in \boldsymbol{\alpha}_K : Z_{\alpha+1/2} + \epsilon_{top} < \min_{\boldsymbol{x} \in K} \zeta^0(\boldsymbol{x}) \right\}, \qquad \alpha_{top,K} = \min \boldsymbol{\alpha}_{active,K} \tag{32}$$

with $\boldsymbol{\alpha}_K = \left\{ 1, ..., N_K \right\}$. Due to the staggering of the grid, it is convenient to define also at each node:

$$\boldsymbol{\alpha}_{active,i} = \left\{ \alpha \in \boldsymbol{\alpha}_i : Z_{\alpha+1/2} + \epsilon_{top} < \zeta_i^0 \right\}, \qquad \alpha_{top,i} = \min \boldsymbol{\alpha}_{active,i} \tag{33}$$

with $\boldsymbol{\alpha}_i = \left\{ 1, ..., N_i \right\}$. The parameter $\epsilon_{top}$ is a small positive constant that fixes the minimum allowable depth for a surface layer to exist. Below this threshold, the layer is removed. We have fixed it as $\epsilon_{top} = 0.2 \Delta Z_\alpha$. It turns out that this parameter

is quite important since it avoids the presence of very small layers. Such layers can lead to a restrictive time step due to the explicit discretization of vertical advection terms. In Figure 5 we illustrate the spatial variation of the top layer index for a one-dimensional example.

## 4.1 Vertical grid movement

We evolve the discrete multilayer shallow water equations with the semi-implicit finite element method detailed in Section 3.

The vertical vectors/matrices are restricted to the layers with active index. Moreover, to account for the movement of the surface layers, the layer thickness is updated as follows:

- we identify the indices associated with the layers that, locally, undergo a deformation. They are defined as the layers of the reference grid whose top interface is above the free surface or by the set of indices:

$$\boldsymbol{\alpha}_{mov,i} = \left\{ \alpha \in \boldsymbol{\alpha}_i : Z_{\alpha-1/2} + \epsilon_{mov} > \zeta_i^{n+1} \right\} \tag{34}$$

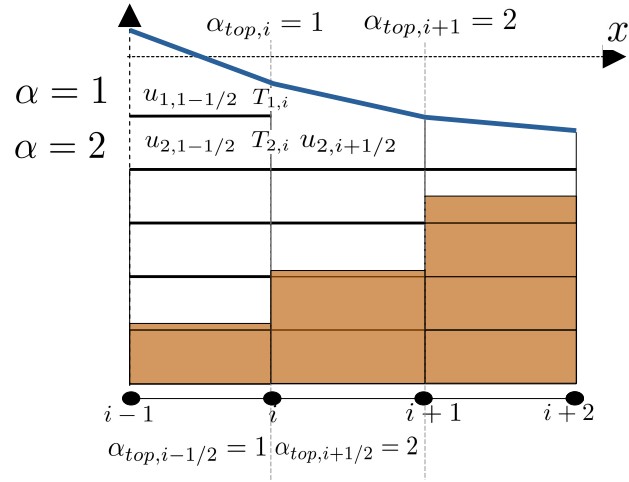

**Figure 5.** This one-dimensional example shows the grid for the $z$-surface-adaptive layers. Elemental surface layer indices are shown on the bottom, nodal surface layer indices are shown on the top.

$\epsilon_{mov}$ is a small and positive constant that we have added. Below this threshold, the vertical grid movement is deployed. As seen for $\epsilon_{top}$, it avoids the presence of very small layers that can be dangerous from a numerical point of view. The bottom-most layer is denoted by $N_{mov,i} = \max \boldsymbol{\alpha}_{mov,i}$. The depth of the moving layers is:

$$z_{mov,i} = \max \left( Z_{N_{mov,i}+1/2}, -z_{b,i} \right)$$

– we compute the new layer thickness after a local grid deformation that absorbs the free surface movement. To move the interfaces of the layers contained in the set, we use the generalized coordinates transformation (1) which takes the form:

$$z_{\alpha+1/2,i}^{n+1} = \zeta_i^{n+1} + S_{\alpha+1/2,i} \left( \zeta_i^{n+1} + z_{mov,i} \right) \tag{35}$$

with $S_{\alpha+1/2,i}$ a stretching function. Then, the nodal layer thickness reads:

$$h_{\alpha,i}^{n+1} = l_{\alpha,i} \left( \zeta_i^{n+1} + z_{mov,i} \right), \qquad \alpha = \alpha_{top,i}, ..., N_{mov,i} \tag{36}$$

For the proportionality coefficients, we have tried different definitions allowing a smooth movement on the interfaces between the time steps, without experiencing any major impact on the results. For simplicity we have thus implemented a $z-$star definition $l_{\alpha,i} = \frac{\Delta Z_\alpha}{z_{mov,i}}$, see Section (2).

This is shown in Figure 4, first and second columns. The new layer thickness is used in the update of the tracers, equation (31). We stress the fact that the vertical configuration is taken constant, i.e. the number of layers at each element remains constant during the time stepping of the discharges and of the tracers.

## 4.2 Removal/Insertion of surface layers

Then we perform the insertion/removal operation based on:

- An update of the active layers and of the top layer index by re-evaluating (32) and (33) with the new free surface elevation $\zeta^{n+1}$. We get the new top layer indices $\alpha_{top,K}^{n+1}$ and $\alpha_{top,i}^{n+1}$

- Once we have identified the index that should be inserted/removed in the active set, we proceed with the collapse/expansion of the surface boxes. A conservative remap step is necessary to pass the unknowns from the old vertical grid to the new one.

We use the tilde $\widetilde{T}_\alpha^{n+1}$ to distinguish a generic layer variable (the tracer in this case) remapped onto the new grid from the solution time stepped on the old grid $T_\alpha^{n+1}$. The remapped value is the solution of the following advection equation integrated over the layer thickness:

$$\frac{\partial \widetilde{h}_\alpha \widetilde{T}_\alpha}{\partial t} = \left[ \sigma^{top} \widetilde{T} \right]_{\alpha+1/2}^{\alpha-1/2} \tag{37}$$

with an upwind flux:

$$\sigma_{\alpha-1/2}^{top} T_{\alpha-1/2}^{n+1} = \left( \sigma_{\alpha-1/2}^{top} \right)^+ T_\alpha^{n+1} + \left( \sigma_{\alpha-1/2}^{top} \right)^- T_{\alpha-1}^{n+1} \tag{38}$$

We consider the discrete case. After integration on the dual cell and with a simple forward Euler time stepping (with initial condition $T_\alpha^{n+1}$) we have:

$$\widetilde{h}_{\alpha,i}^{n+1} \widetilde{T}_{\alpha,i}^{n+1} = h_{\alpha,i}^{n+1} T_{\alpha,i}^{n+1} + \Delta t \left( \sigma_{\alpha-1/2,i}^{top} T_{\alpha-1/2,i}^{n+1} - \sigma_{\alpha+1/2,i}^{top} T_{\alpha+1/2,i}^{n+1} \right) \tag{39}$$

with the new nodal layer thickness:

$$\widetilde{h}_{\alpha,i}^{n+1} = \widetilde{z}_{\alpha-1/2,i}^{n+1} - \widetilde{z}_{\alpha+1/2,i}^{n+1}$$

In the case of an element removal ($\alpha_{top,i}^{n+1} > \alpha_{top,i}^n$), we identify the layer that should disappear and we proceed with a collapse of the lower interface to the upper one. For $\alpha = \alpha_{top,i}^n, ..., \alpha_{top,i}^{n+1}$, the discrete remap (39) with (38) reduces trivially to transfer the depth-integrated tracer that belongs to the removed layers to the upper active layer. In the case of an element insertion ($\alpha_{top,i}^{n+1} < \alpha_{top,i}^n$), we identify the layer that should appear and we expand the interface. Then the remap for $\alpha = \alpha_{top,i}^{n+1}, ..., \alpha_{top,i}^n$ reduces to distribute the depth-integrated variable across the existing and inserted layers with a weighted average. This is shown in Figure 4, third and fourth columns. All the unknowns must be remapped. For the discharges, that are defined on the elements, (37) should be integrated on the element. This completes the time step.

## 4.3 Connection to $z-$star

We have a small parameter $\epsilon_{mov}$ to fix. It is convenient to express this constant as a percentage of the reference layer thickness, $\epsilon_{mov} = r_{mov} \Delta Z_\alpha$ in the relationship (34). In order to obtain the $z-$surface-adaptive grid we have chosen $r_{mov} \leq r_{top}$, in practice we have set $r_{mov} = 0.15$. The grid deformation is localized to the free surface. As long as the surface fluid boxes are deformed, they are recognized as too small and immediately removed in the grid topology step. This implies working, at the next time step, with $z-$layers having all the interfaces aligned to the geopotentials.

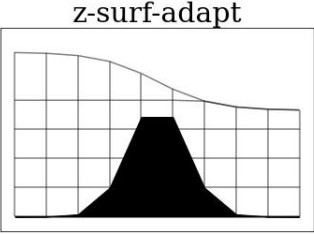
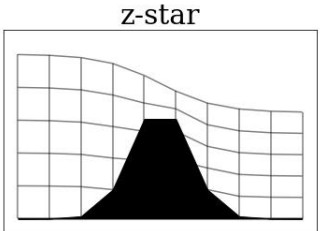
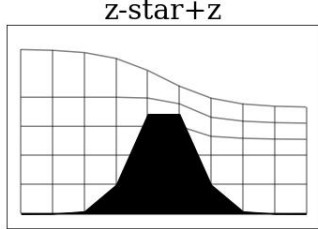

z-surf-adapt           z-star           z-star+z

**Figure 6.** The different vertical grids outlined in Section 4.3.

Interestingly we can obtain other grids by increasing $r_{mov}$. We define:

$$R_\alpha = \frac{\zeta_{max} - Z_{\alpha-1/2}}{\Delta Z_\alpha} \tag{40}$$

with $\zeta_{max} \geq \max_{\boldsymbol{x},t} \zeta(\boldsymbol{x},t)$ an estimate of the maximum free surface height during the simulation. We get:

- $z-$star if $r_{mov} \geq R_N$ and no insertion/removal. The whole water column is subjected to the grid movement while the number of layers does not change.

- $z-$star+z if $r_{mov} = R_M$ and no insertion/removal. The upper part of the water column, at minimum $M$ layers, is subjected to the grid movement while the lower part is fixed.

Figure 6 shows a sketch of the different possibilities. Tuning properly $r_{mov}$ we will compare the newly developed $z-$surface adaptive layers against $z-$star.

## 5   Advection with spatially variable number of layers

We have used an approach where the grid topology does not change during the time step of the conserved variables, i.e. the numerical scheme of Section 3 works on the deforming grid of Section 4.1, with a *temporally* constant number of layers between $t^n$ and $t^{n+1}$. However, in the previous time step, a layer insertion/removal may occur (to remove very thin surface layers or to split a thicker layer) on a certain element and not on its neighbors. This results in a vertical discretization with a *spatially* variable number of layers, see Figure 7, which slightly complicates the treatment of advection terms, see on this topic Bonaventura et al. (2018).

Consider the one dimensional example in Figure 7, where two contiguous elements with different top-layer index $\alpha_{top,i+1/2} > \alpha_{top,i-1/2}$ exist. In correspondence with node $i$ a change of the element top layer index takes place. Borrowing the vocabulary from the literature on non-conformal meshes, we have a vertical edge with a hanging point. We call hanging layer, a layer for which at least one interface ends with a hanging point. The boxes that have vertical edges across which the element top-layer index varies deserve special treatment. In our case, with only insertion/removal of surface layers, we can easily flag such boxes by checking, for each element, that the nodal top layer index is different from the elemental one. The elements of the grid with

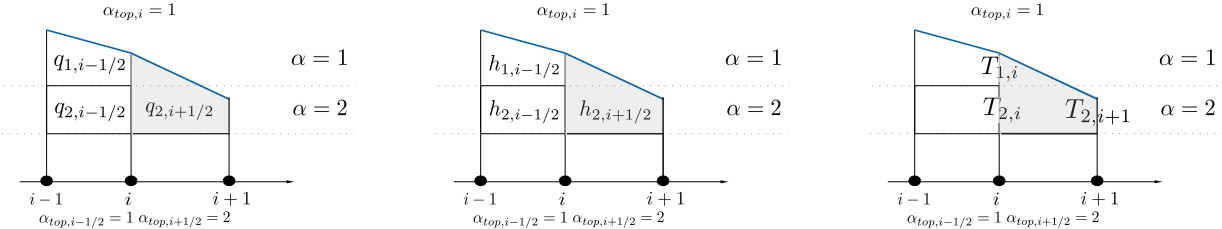

**Figure 7.** Non-conformal box for the one-dimensional case. The non-conformal box is in grey. Discharges, layer thickness and tracers are shown.

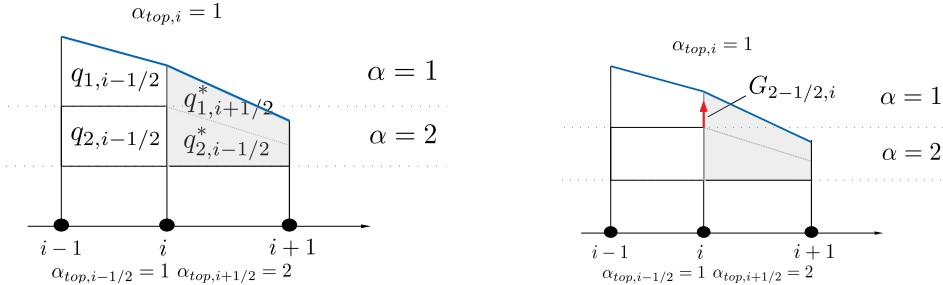

**Figure 8.** Treatment of non-conformal box for the one-dimensional case. Left: splitting with fictitious layers. Right: the mass-transfer function $G_{1+1/2,i}$ at hanging point is represented by a red arrow.

a non-conformal surface box are indicated by an asterisk:

$$\text{if } \alpha_{min,K} < \alpha_{top,K} \quad \text{then} \quad K = K^*$$

with $\alpha_{min,K} = \min_{j \in K} \alpha_{top,j}$. Then the boxes called hereinafter for simplicity "non-conformal" can be identified by the pair of indices $(\alpha_{top,K}, K^*)$. Since both mass and tracer fluxes need communication with the neighbors' boxes, they have to be treated differently. Moreover, for the tracer discrete update, we have to take care of preserving the constancy property.

In case of a non-conformal box we proceed as follows. We split the box vertically in fictitious layers through planar interfaces passing through the hanging points of non-conformal edges and some fraction of the conformal edge length, see Figure 8, left panel. From this geometrical configuration we compute the element layer thickness $h_{\alpha,K}^*$ for the fictitious layers. Then we distribute the discharge of the top layer among the fictitious layers, for $\alpha = \alpha_{min,K}, ..., \alpha_{top,K}$:

$$\boldsymbol{q}_{\alpha,K}^* = l_{\alpha,K}^* \, \boldsymbol{q}_{\alpha_{top,K},K} \tag{41}$$

with $l_{\alpha,K}^* = \dfrac{h_{\alpha,K}^*}{h_{\alpha_{top,K}K}}$. These values are used to complete both mass and tracer fluxes for the missing layers of non-conformal boxes. We consider the case of a non-conformal box $(\alpha_{top,K}, K)$ with node $i \in K$, as illustrated in one dimension in Figure 7.

After the splitting (41), the mass-flux term (only the $x-$component shown) reads, for $\alpha = \alpha_{top,i}, ..., \alpha_{top,K}$:

$$\int_K \frac{\partial \varphi_i}{\partial x} q_\alpha^* d\boldsymbol{x} = a_{iK} c_{\alpha,i}^* q_{\alpha_{top,K},K} \tag{42}$$

with:

$$c_{\alpha,i}^* = \begin{cases} \sum_{\beta=\alpha_{top,i}}^{\alpha_{min,K}} l_{\beta,K}^* & \text{if} \quad \alpha = \alpha_{top,i} \quad \text{and} \quad \alpha_{min,K} < \alpha_{top,i} \\ l_{\alpha,K}^* & \text{otherwise (hanging layer)} \end{cases} \tag{43}$$

where the two cases account for the contribution of element $K$ to nodes with or without hanging layers, respectively node $i$ and $i+1$ in Figure 8. Such contribution from the non-conformal box is added to the mass-flux term in the layerwise mass equation. It allows to compute the mass-transfer function at the hanging points $G_{\alpha-1/2,i}^{n+1}$ for $\alpha = \alpha_{top,i}, ... \alpha_{top,K}$ as shown in Figure 8, right panel. One can check that this treatment is mass-conserving. Summing the mass-transfer function for all the layers, even in presence of non-conformal boxes, still yields the discrete mass equation (27).

The horizontal advection scheme (29) on the non-conformal box can be applied straightforwardly to the fictitious layers. Then, the numerical flux in non-conformal boxes reads for $\alpha = \alpha_{top,i}, ..., \alpha_{top,K}$:

$$\widehat{H}_\alpha = \begin{cases} \sum_{\beta=\alpha_{top,i}}^{\alpha_{min,K}} l_{\beta,K}^* \widehat{H}_{\alpha_{top,K}}(T_{\beta^*,i}, T_{\beta^*,j}) & \text{if } \alpha = \alpha_{top,i} \text{ and } \alpha_{min,K} < \alpha_{top,i} \\ l_{\alpha,K}^* \widehat{H}_{\alpha_{top,K}}(T_{\alpha^*,i}, T_{\alpha^*,j}) & \text{otherwise (hanging layer)} \end{cases} \tag{44}$$

Again we have separated the cases of a node with or without hanging layers. Note that the subscript $\alpha^* = \max(\alpha, \alpha_{top,j})$ avoids selecting tracer values in removed layers. In the Appendix we show that, when a constant tracer is imposed, the horizontal tracer flux reduces to the mass flux even in the case of a non-conformal box.

# 6 Numerical tests

The tests have been run with implicitness parameter $\theta = 0.5$. We will check discrete mass-conservation at $t^{n+1}$ by computing the following relative volume error for the dual cell area, which results from the sum of (28) from $N_i$ to $\alpha_{top,i}$:

$$e_i^{n+1} = \Delta t \left| \sum_{\alpha=N_i}^{\alpha_{top,i}} |C_{\alpha,i}| G_{\alpha-1/2,i}^{n+1} \right|, \qquad e^{n+1} = \max_{i \in \mathcal{T}} \left( \frac{e_i^{n+1}}{\sum_{\alpha=N_i}^{\alpha_{top,i}} |C_{\alpha,i}| h_{\alpha,i}} \right)$$

To quantify the tracer constancy error, we use the $L^1-$norm:

$$e^{n+1} = \frac{\sum_{\alpha,i} |C_{\alpha,i}| h_{\alpha,i}^{n+1} |T_{\alpha,i}^{n+1} - T_0|}{\sum_{\alpha,i} |C_{\alpha,i}| h_{\alpha,i}^{n+1} T_0}$$

with $T_0$ the initial tracer value.

## 6.1 Impulsive Wave

As the first test, we check the accuracy of the $z-$surface-adaptive layers with an increasing vertical resolution. We use a closed basin $[-5\,\mathrm{m}, 5\,\mathrm{m}] \times [-5\,\mathrm{m}, 5\,\mathrm{m}]$ with constant depth $z_b = 1\,\mathrm{m}$. The basin is initially at rest and the free surface is perturbed by the following Gaussian hump:

$$\zeta(\boldsymbol{x}, t=0) = A \exp(-r^2/\tau)$$

with $A = 0.5\,\mathrm{m}$, $\tau = 0.5\,\mathrm{m}^2$ and $r = \sqrt{x^2 + y^2}$. A constant passive tracer is prescribed on the background and such a constant state should be preserved along the simulation. The mesh has a uniform horizontal element size of $\mathrm{h}_K = 0.25\,\mathrm{m}$. We compare different vertical resolutions with variable layer thickness. The coarsest grid has three layers: a first surface layer with thickness of $\Delta Z_1 = 0.2\,\mathrm{m}$, the second and the third layers have thicknesses of $\Delta Z_2 = \Delta Z_3 = 0.4\,\mathrm{m}$. The other vertical grids are obtained by halving each of these layers. The finest grid has 24 layers with minimum layer thickness at the surface of $\Delta Z_1 = 0.025\,\mathrm{m}$.

Without bottom/surface forcing, if the initial velocities are constant over the layers, they must remain barotropic and equal to the depth-integrated velocities of the shallow water equations (1-layer case). Of course, this is not a property of the $z-$layers (but the scheme should converge to a barotropic solution refining the resolution). It is however desirable that the results of 2d and 3d models are similar for the typical resolution of an ocean simulation (Kleptsova et al., 2010). The 1-layer discrete solution is considered here as a reference solution against which we compare our implementation of the $z-$layers. The coarse grid with 3-layer is also used for comparison since the free surface is contained in the first layer and no insertion/removal is necessary. For the 24-layer grid, up to six layers are progressively removed (and then re-inserted). In Figure 9, all resolutions show a good agreement for both the water level and the barotropic velocity. We can check some conservation properties of the scheme. As usual for such an adaptation strategy, mass is conserved up to machine precision (SHYFEM is coded in single-precision). This is what we check in Figure 10, left panel. Except for a small additional noise associated with the insertion/removal operations, no significant source of mass error is present with respect to the 3-layer case. Tracer constancy, as expected, is also preserved up to machine precision, see Figure 10, right panel.

## 6.2 1-d tidal flow in a sloping channel

Coastal applications include extensive intertidal flats. As with many ocean models, SHYFEM handles wetting and drying processes in a simplified manner, applying special treatments in dry cells. An extrapolation algorithm for the free surface is used to track the shoreline and identify dry and wet elements. Then, the dry elements are taken out from the semi-implicit update and are treated in a mass-conserving manner as described in Umgiesser (2022). The test that we propose, presented in Oey (2005), is a benchmark for wetting/drying algorithms used in ocean models. The domain consists of a 1d sloping channel that ranges from $x = 0$ at the landward end to $x = L$ at the seaward boundary, with $L = 25\,\mathrm{km}$. The bathymetry is represented by the following function $z_b(x) = H_0/L\,x$ and $H_0 = 10\,\mathrm{m}$. The horizontal element size is uniform and equal to $\mathrm{h}_K = 250\,\mathrm{m}$. A periodic water level is imposed at the seaward boundary as $\zeta(t) = A(1 - \sin\left(\frac{2\pi t}{T}\right))$ with amplitude $A = 10\,\mathrm{m}$, period $T = 1\,\mathrm{day}$ and the time $t$ ranging from 0 to $0.5\,\mathrm{day}$. At the beginning of the simulation, the channel is dry. Typically

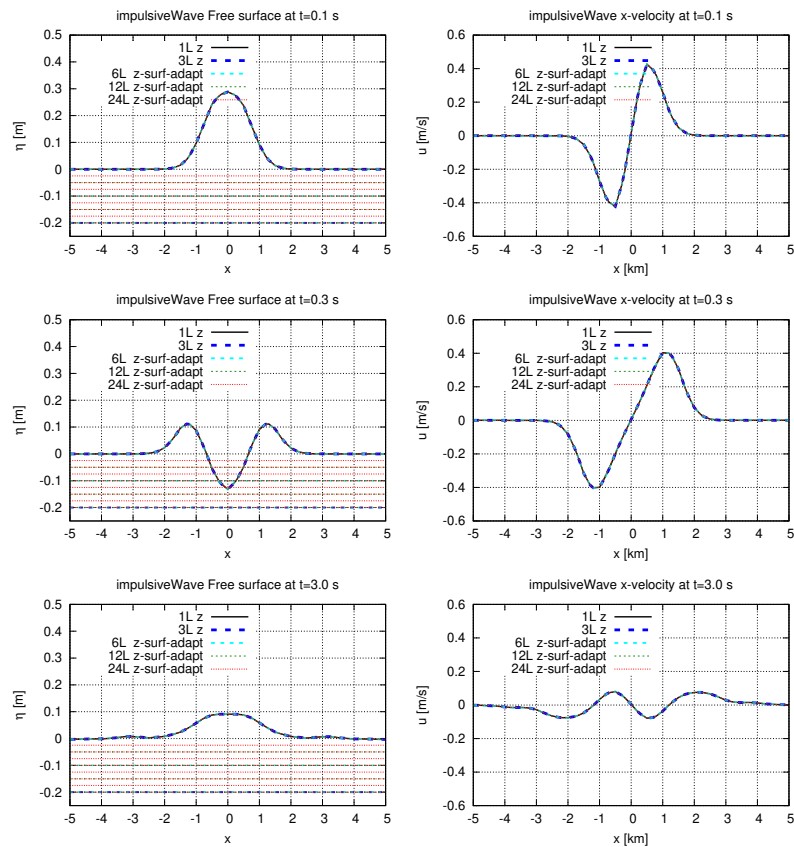

**Figure 9.** Impulsive Wave. Comparison of the free surface elevation and barotropic velocity at different time instants. Vertical grids with different resolutions are compared. For each grid the reference interfaces $Z_{\alpha+1/2}$ are traced with dashed lines. In the regions where the free surface crosses the interface $Z_{\alpha+1/2}$ it means that the layer $\alpha$ locally has been removed from the computation.

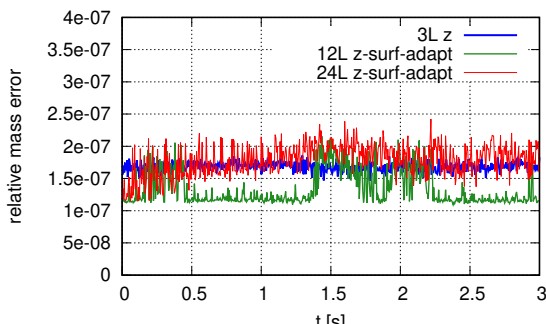

| Number of Layers | Relative tracer error |
|---|---|
| 3L $z-$surf-adapt | 3.585e-08 |
| 6L $z-$surf-adapt | 1.347e-08 |
| 12L $z-$surf-adapt | 6.744e-09 |
| 24L $z-$surf-adapt | 2.954e-09 |

**Figure 10.** Impulsive Wave. Left: relative mass conservation error for the dual cell. Right: relative tracer constancy error at the final time $t = 3\,s$.

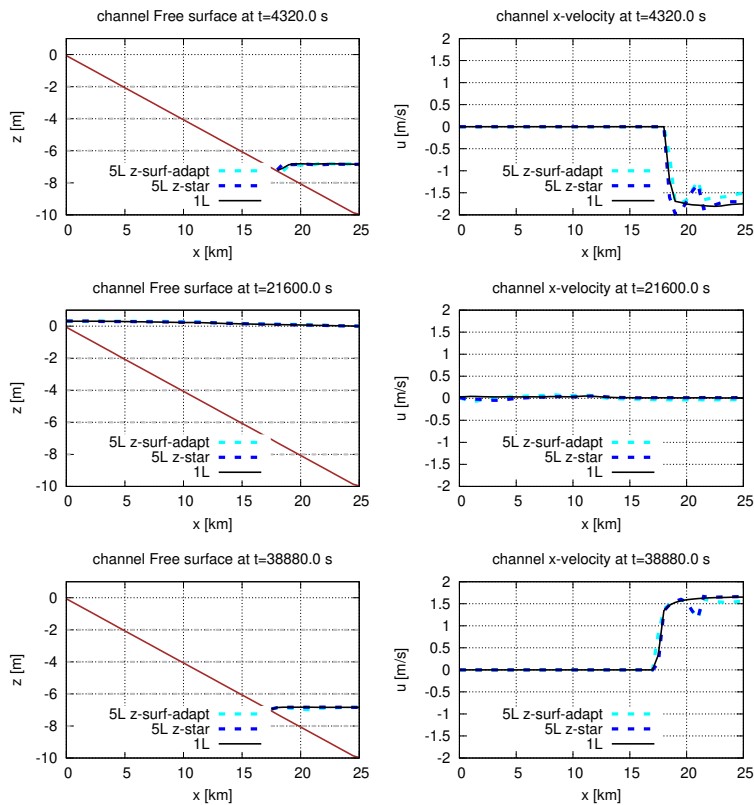

**Figure 11.** 1d tidal channel flow. Comparison between the 1-layer and 5-layer runs. Left: free surface elevation. Right: barotropic velocity. Dashed grey lines represent the reference interfaces $Z_{\alpha+1/2}$. In the regions where the free surface crosses the interface $Z_{\alpha+1/2}$ it means that the layer $\alpha$ locally has been removed from the computation.

this test is run with 1-layer models (Warner et al., 2013). Here we use the 1-layer solution (1L) as a reference and we test the
510 5-layer with surface-adaptation and the 5-layer with $z-$star. In the 5L $z-$surface-adaptive simulation, only one layer is present at the beginning of the simulation and then, as the free surface is tilted by the boundary signal, more levels are inserted and then removed during the drying phase. With $z-$star instead, the number of layers remains constant over time.

In Figure 11 we check the along-channel solution profiles. Despite the different vertical resolution in the wet/dry and dry regions for the 5L $z-$surface-adaptive and 5L $z-$star simulations, a quite good agreement is observed for the free surface.
Larger differences are found for the barotropic velocity where both the 5-layer simulations appear noisier at the wet/dry interface. In Figure 12, left panel, we check volume conservation for this case which involves an uneven bathymetry and wetting/drying. Although in correspondence of wet/dry nodes the relative volume error is much larger, we can verify that the $z-$surface adaptive has the same level of relative error of $z$-star, which we accept to be within the round-off errors. The same argument applies to the error for the tracer constancy.

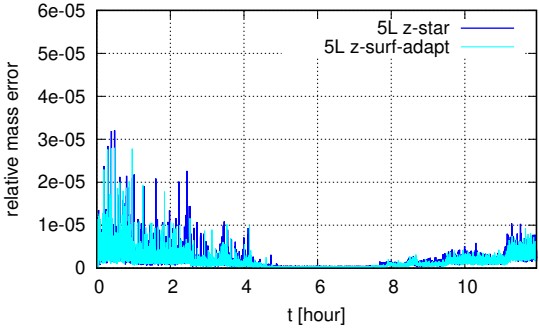

| Number of Layers | Relative racer error |
|---|---|
| 5L $z-$surf-adapt | 1.195e-05 |
| 5L $z-$star | 7.415e-06 |

**Figure 12.** 1d tidal channel flow. Left: relative mass conservation error for the dual cell. Right: relative tracer constancy error at the final time.

### 6.3 Po delta idealized test

We test the different $z-$layers in a realistic coastal environment forced by the tidal oscillation: the Po delta. We study both the river plume and the penetration of the salt water into the river branches. The reproduction of such phenomena for numerical models is a very delicate issue. Specifically, spurious mixing related to the horizontal and vertical numerical fluxes, the vertical grid and the time-stepping can destroy stratification and frontal characteristics, potentially modifying the plume dynamics (Fofonova et al., 2021). In this discussion we solely focus on the impact of the vertical discretization: the resolution at the surface and the comparison between the $z-$surface adaptive with fixed interfaces and $z-$star with moving interfaces.

The vertical eddy viscosity and the vertical tracer eddy diffusivity are computed with the turbulence module GOTM (Buchard et al.). The bottom friction is fixed to $C_F = 0.002$. Because of their fundamental role in the plume dynamics, two more terms have been added to the multilayer shallow water model of Section 2: the Coriolis force which is time stepped with an implicitness parameter of $0.5$ and a horizontal diffusion term for the salinity equation, treated explicitly. The horizontal viscosity is taken as the Smagorinsky eddy viscosity. The sea boundary is forced with a semi-diurnal tidal signal with amplitude $0.4\,\mathrm{m}$ and period $12$ hours. The salinity at the sea boundary is constant and fixed to $38\,\mathrm{PSU}$. A weak freshwater flow with a discharge of $500\,\mathrm{m}^3 s^{-1}$, which is characteristic of the summer season, is enforced at the Pontelagoscuro river boundary. The initial solution corresponds to water at rest and salinity equal to the boundary value of $38\,\mathrm{PSU}$. The simulation lasts one month, after which the salinity shows a periodic behavior modulated by the tidal cycle.

The computational domain encompasses the entire river network of the delta, stretching from Pontelagoscuro to the sea, including all delta lagoons, as well as a portion of the adjacent shelf sea (Bellafiore et al., 2021). Horizontal resolution ranges from $\mathrm{h}_K = 2\,\mathrm{km}$ at the sea boundary, to $\mathrm{h}_K = 100\,\mathrm{m}$ in the inner shelf close to the lagoons and river branches, and to $\mathrm{h}_K = 50\,\mathrm{m}$ in the inner delta system. The horizontal grid, composed of 38884 nodes and 69364 elements, is in Figure 13. We consider two vertical resolutions, one with $N = 24$ layers and one with $N = 27$ layers. The deeper part (from the bottom to $Z = -1\,\mathrm{m}$) is equal for the two grids and it is composed of 23 levels with variable thicknesses from $\Delta Z = 0.5\,\mathrm{m}$ near the surface up to $\Delta Z_N = 4\,\mathrm{m}$ for the last layer. The resolution of the upper part of the water column differs: the 24-layer grid has one layer with

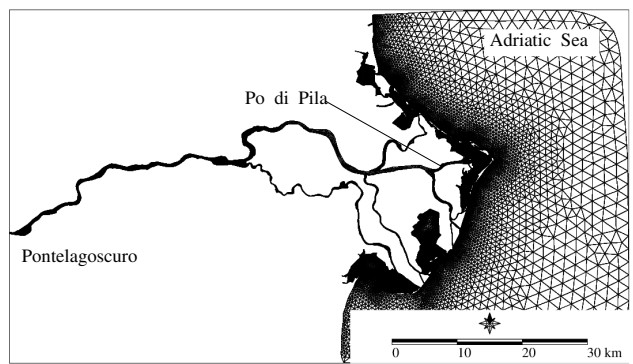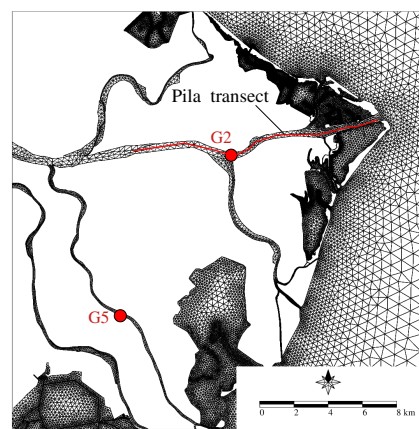

**Figure 13.** Po river. Left: horizontal grid. Right: zoom of the horizontal grid with tidal stations and the transect in the Pila branch.

$\Delta Z_1 = 1\,\mathrm{m}$. This choice avoids the drying of the first layer. The 27-layer grid, in the upper part, has 4 layers with constant thickness, $\Delta Z_1 = \Delta Z_2 = \Delta Z_3 = \Delta Z_4 = 0.25\,\mathrm{m}$. Three simulations have been performed: one with 24 standard $z-$layers (24L $z$), one with 27 $z-$surface-adaptive layers (27L $z-$surf-adapt) and one with 27 $z-$star layers (27L $z-$star).

Given the fine vertical resolution at the surface and the tidal amplitude of $0.4\,\mathrm{m}$, the 27L $z-$surf-adapt simulation should undergo extensive insertion/removal of the surface fluid boxes. In the top left picture of Figure 14 we have reported the time evolution of the number of boxes inserted and removed during two tidal periods. Almost 4000 surface boxes happened to be inserted or removed in a single time step. As it is customary we have reported mass conservation and tracer constancy error in Figure 14. These figures refer to a shorter simulation that lasted 4 days with a constant salinity obtained by imposing the river salinity equal to the interior one.

To diagnose the river plume we look at the minimum surface salinity during the simulation. From Figure 15, it is clear that both the 27-layer simulations allow a stronger gravitational circulation with a more extended freshwater plume. Also, the opposite bottom circulation penetrates more upstream, with stronger salinity recorded at the stations G2 and G5, as shown in Figure 16. To inspect the extension of the saltwater intrusion we have extracted a section of the salinity field in the Pila branch when saltwater reaches the maximum extent, during a flood tide. This is shown in Figure 17. The higher resolution at the surface also allows to capture some small-scale internal structures that are present under the surface. The $z$-surface adaptive simulation exhibits a stronger plume and a more extended salt wedge as well as a sharper surface structure. Similar results can be observed in a recent work (Verri et al., 2023), where standard $z$-layers and $z$-star are compared for an analogous river plume experiment. A possible explanation could be related to the fact that, due to the strong internal motion, the vertical velocity is not in phase with the time derivative of the free surface and it may have opposite sign with respect to the grid velocity. For this particular case, the $z$-star mass-transfer function (11) could be larger than the vertical velocity. In turn, this can be related to a larger multiplicative constant in the truncation error associated with the vertical advection scheme.

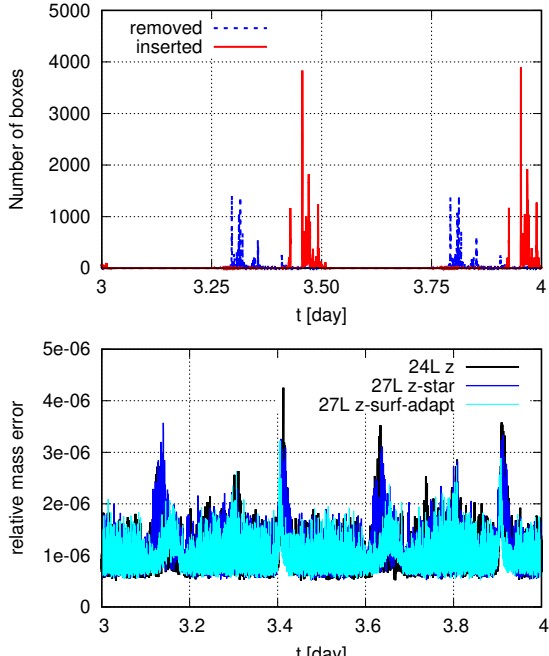

| Number of Layers | Relative tracer error |
|---|---|
| 24L $z$ | 2.011e-07 |
| 27L $z-$surf-adapt | 5.226e-07 |
| 27L $z-$star | 2.495e-07 |

**Figure 14.** Po river. Top left: time evolution of the total number of layers inserted and removed per time step for the 24L $z-$surf-adapt simulation. Bottom left: relative mass conservation error for the dual cell. Right: relative tracer constancy error after 4 days.

All the tests have been accomplished with a serial run. We report the CPU time of the serial simulations which have been run on a modern workstation with a AMD EPYC 7643 Processor : 2073005 s (24L $z-$star), 1998969 s (24L $z-$surf-adapt) showing an overhead of around 3.6% for the insertion/removal operations. Although we have not covered parallel implementation aspects, we mention that the algorithm (grid movement, insertion/removal) mainly operates on the vertical grid, and the parallel execution of these tasks should not encounter any issues. The stencil of the numerical scheme is not enlarged with respect to the standard method. However some variables have been introduced only for the insertion/removal operations. This is the case of the nodal top layer index which must be exchanged between the domains.

## 7 Conclusions

In this work, we have studied the performances of multilayer shallow water models based on $z-$layers for the simulation of free surface coastal flows. We have investigated a well-known issue of $z-$layers when incorporating the free surface: the limitation on the resolution of the surface layer thickness. We have proposed a flexible algorithm based on a vertical mesh adaptation to the tidal oscillation called $z-surface-adaptive$. With a dynamic insertion and removal of surface layers, the grid at the new timestep is aligned to the geopotentials, reducing the pressure gradient error. Thanks to a two-step procedure (vertical grid movement of surface layers followed by the insertion/removal operations), we have been able to evolve the multilayer model on a grid with a temporally constant number of layers in the time step which allowed a simple implementation. Moreover, this

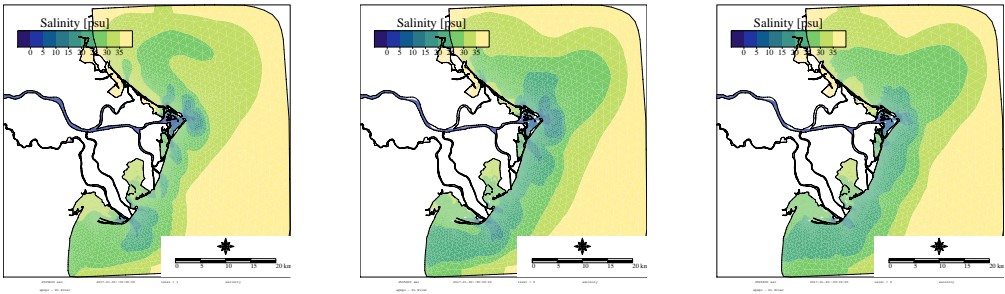

**Figure 15.** Po river. Minimum of the surface salinity (for the 24-layer grid the minimum is computed at the first layer, for the 27-layer grid at the second layer). Left: 24L $z$. Middle: 27L $z-$surf-adapt. Right: 27L $z-$star.

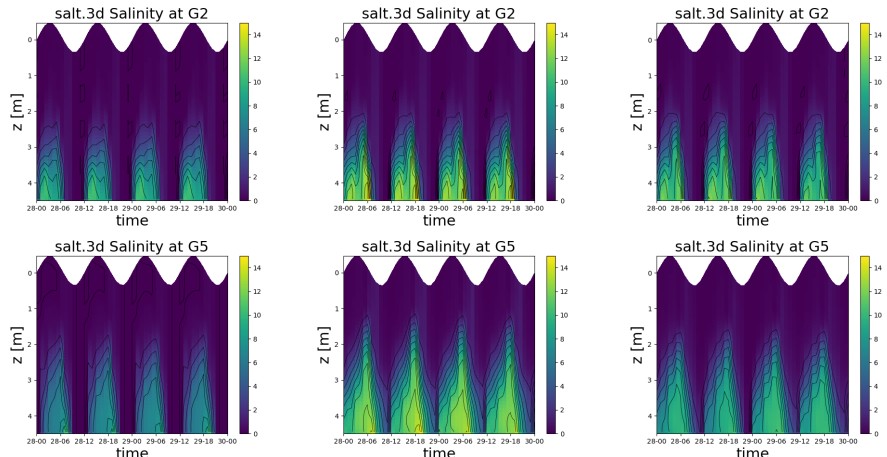

**Figure 16.** Po river. Salinity profile at G2 (top) and G5 (bottom). Left: 24L $z$. Middle: 27L $z-$surf-adapt. Right: 27L $z-$star.

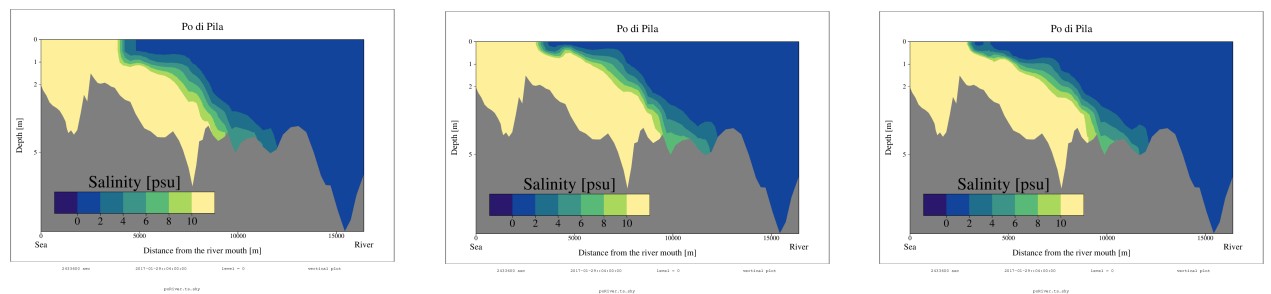

**Figure 17.** Salinity section along the Pila branch during the flood tide of day 29 16:00. Left: 24L $z$. Middle: 27L $z-$surf-adapt. Right: 27L $z-$star.

leads to a consistency, at a discrete level, of the tracer equation with the continuity equation as well as to a simple verification of mass-conservation. As a particular case, the algorithm can be reduced to the popular $z-$star.

Without the limitation on the surface resolution, we have been able to compare the $z-$layers with insertion/removal (surface-adaptive) against $z-$star for typical coastal applications of semi-enclosed shallow seas with a tidal signal imposed at the openings and wetting/drying at intertidal flats. The comparison has been carried out with idealized and realistic numerical experiments. We show that $z-$surface-adaptive layers can be used to simulate wetting and drying without a significant loss of accuracy with respect to $z$-star. We found that $z$-layers and $z-$star exhibit differences when simulating large, low frequency internal motions combined with a barotropic tide, such as the gravitational circulation in the Po Delta. These differences deserve further investigation. We speculate that for such cases, keeping $z$-layers may be convenient to reduce truncation errors in the computation of both the internal pressure gradient term and the vertical advection terms.

We conclude by mentioning that the overhead related to insertion/removal operation should be further assessed in realistic applications. With the actual implementation of the $z$-surface adaptive layers, we have experienced some stability issues in the computation of the tracers. This occurred for non-conformal boxes undergoing wetting/drying and it is under current investigation. We are trying a simpler treatment of the non-conformal surface boxes as in Bonaventura et al. (2018).

*Code and data availability.* The SHYFEM hydrodynamic model is open source (GNU General Public License as published by the Free Software Foundation) and freely available through GitHub at https://github.com/SHYFEM-model. The current developments have been implemented in a branch of the SHYFEM code that can be accessed from Zenodo (Arpaia, 2023, https://doi.org/10.5281/zenodo.8356398). Configuration files and data used to run each test case are also available at the same Zenodo repository.

## Appendix A: Tracer constancy

We start with the case without non-conformal boxes. We impose a constant tracer vector $\boldsymbol{T}_i = \boldsymbol{1}$ in the discrete tracer equation (31). Each row reduces to:

$$|C_{\alpha,i}|h_{\alpha,i}^{n+1} \quad = \quad |C_{\alpha,i}|h_{\alpha,i}^n + \Delta t \, f_{\alpha,i}^n$$

with

$$f_{\alpha,i}^n = -\sum_{K \in \mathcal{D}_{\alpha i}} \sum_{j \in K, j \neq i} \widehat{H}_\alpha (1,1) + \left( |C_{\alpha,i}| G_{\alpha-1/2,i}^{n+1} - |C_{\alpha+1,i}| G_{\alpha+1/2,i}^{n+1} \right)$$

Using, first, the numerical flux consistency $\widehat{H}_\alpha (1,1) = \boldsymbol{q}_\alpha^{n+\theta} \cdot \boldsymbol{n}_{ij}^K$ and then the relationship between the element normals and the dual cell ones (19):

$$\sum_{K \in \mathcal{D}_{\alpha i}} \sum_{j \in K, j \neq i} \widehat{H}_\alpha (1,1) \quad = \quad \sum_{K \in \mathcal{D}_{\alpha i}} \sum_{j \in K, j \neq} \boldsymbol{q}_\alpha^{n+\theta} \cdot \boldsymbol{n}_{ij}^K = -\sum_{K \in \mathcal{D}_{\alpha i}} \boldsymbol{q}_\alpha^{n+\theta} \cdot \frac{\boldsymbol{n}_i^K}{2}$$

$$= \quad -\sum_{K \in \mathcal{D}_{\alpha i}} \left( a_{iK}^x q_{\alpha,K}^{x,n+\theta} + a_{iK}^y q_{\alpha,K}^{y,n+\theta} \right)$$

In the last step we have used the fact the for piecewise linear basis functions we have $\frac{n_i^K}{2} = |K| \left. \nabla \varphi_i \right|_K$. For each element in the subset $\mathcal{D}_{\alpha,i}$, the horizontal tracer flux has been reduced to the mass flux. We can write the discrete tracer update:

$$|C_{\alpha,i}| \frac{\Delta h_{\alpha,i}}{\Delta t} = \sum_{K \in \mathcal{D}_{\alpha i}} \left( a_{iK}^x \, q_{\alpha,K}^{x,n+\theta} + a_{iK}^y \, q_{\alpha,K}^{y,n+\theta} \right) + |C_{\alpha,i}| G_{\alpha-1/2,i}^{n+1} - |C_{\alpha+1,i}| G_{\alpha+1/2,i}^{n+1}$$

which corresponds to the discrete layerwise mass equation (28).

In case of a non-conformal box, we have to show that the modified horizontal tracer fluxes still reduces to the mass-fluxes. According to (44), the horizontal tracer fluxes in non-conformal boxes should be computed with:

$$\widehat{H}_\alpha = \begin{cases} \displaystyle\sum_{\beta=\alpha_{top,i}}^{\alpha_{min,K}} l_{\beta,K}^* \, \widehat{H}_{\alpha_{top,K}}(T_{\beta^*,i}, T_{\beta^*,j}) & \text{if } \alpha = \alpha_{top,i} \text{ and } \alpha_{min,K} < \alpha_{top,i} \\[1em] l_{\alpha,K}^* \, \widehat{H}_{\alpha_{top,K}}(T_{\alpha^*,i}, T_{\alpha^*,j}) & \text{otherwise (hanging layer)} \end{cases}$$

For a constant tracer, it can be rewritten for $\alpha = \alpha_{top,i}, ... \alpha_{top,K}$:

$$\widehat{H}_\alpha = c_{\alpha,i}^* \widehat{H}_{\alpha_{top,K}}(1,1)$$

where we have also used the definition (43). Thus:

$$\sum_{j \in K, j \neq i} c_{\alpha,i}^* \widehat{H}_{\alpha_{top,K}}(1,1) = -c_{\alpha,i}^* \left( a_{iK}^x \, q_{\alpha_{top,K},K}^{x,n+\theta} + a_{iK}^y \, q_{\alpha_{top,K},K}^{y,n+\theta} \right)$$

This gives exactly the contribution from non-conformal boxes to the mass-transfer (42).

Finally, the tracer remap (39) preserves the constancy property. It is enough to verify that with a constant solution it reduces to:

$$\widetilde{h}_{\alpha,i}^{n+1} = h_{\alpha,i}^{n+1} + \Delta t \left( \sigma_{\alpha-1/2,i}^{top} - \sigma_{\alpha+1/2,i}^{top} \right)$$

which, thanks to the definition provided in Section 4.2 of grid velocity $\sigma_{\alpha-1/2,i}^{top} = \frac{\widetilde{z}_{\alpha-1/2,i}^{n+1} - z_{\alpha-1/2,i}^{n+1}}{\Delta t}$ and layer thickness $\widetilde{h}_{\alpha,i}^{n+1} = \widetilde{z}_{\alpha-1/2,i}^{n+1} - z_{\alpha+1/2,i}^{n+1}$, is an identity.

*Author contributions.* L. Arpaia: Conceptualization, Methodology, Software, Validation, Writing, Formal analysis. C. Ferrarin: Conceptualization, Methodology, Funding acquisition, Writing, Resources, Validation. M. Bajo: Methodology, Writing. G. Umgiesser: Conceptualization, Methodology, Writing, Software.

*Competing interests.* The authors declare that they have no known competing financial interests or personal relationships that could have appeared to influence the work reported in this paper.

*Acknowledgements.* This work was partially supported by the project AdriaClim (Climate change information, monitoring and management tools for adaptation strategies in Adriatic coastal areas; project ID 10252001) funded by the European Union under the V-A Interreg Italy-

Croatia CBC programme. All the developments presented have been implemented in the Finite Element Model for Coastal Seas SHYFEM (https://github.com/SHYFEM-model/shyfem) developed at the CNR-ISMAR. The authors acknowledge Dr. Debora Bellafiore for the fruitful discussions about the implementation of the present work. The corresponding author expresses his gratitude to Prof. Luca Bonaventura and Dr. Giacomo Capodaglio, the two reviewers, for their valuable comments and feedback that contributed to improve the precision and clarity of the manuscript during the revision process.

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
