# Peer review of "A flexible z-layers approach for the accurate representation of free surface flows in a coastal ocean model (SHYFEM v. 7\_5\_71)"

_Geoscientific Model Development, 2023_

## Referee Comment (RC1)

**Specific comments:**

1) Starting with equations 3-4 the authors introduce an undefined quantity $h$ that seems to play the role of vertical spacing. This quantity appears alongside the quantity $h_\alpha$, which instead is defined on page 4. Later on, quantities $l_\alpha$ are introduced and employed without being defined. Whether these have the same meaning usually implied in the multilayer literature is never stated. This does not allow the reader to understand exactly how the proposed numerical method is formulated and makes the results described in the preprint impossible to reproduce. A mandatory revision is to define properly all quantities that are being introduced and explain clearly their relationship (if any) to analogous quantities introduced in the literature on multi-layer models. Furtermore, in the appendix (line 386) the authors seem to imply that in models for which $h_\alpha = l_\alpha H$ the mass transfer terms across the layers have to be zero. However, this is not true for the (Audusse et al. 2011) paper which apparently constitutes the reference for the present paper nor for the related discretization approach introduced in

   Fernandez-Nieto, E. D., Kone, E. H., Chacon Rebollo, T. (2014). A multilayer method for the hydrostatic Navier-Stokes equations: a particular weak solution. Journal of Scientific Computing, 60, 408-437.

   and employed in (Bonaventura et al. 2018). The authors should clearly specify to which multi-layer formulation they refer and remove any incorrect statements in this respect.

2) The authors devote a significant effort to the important issue of proving that what they call the 'tracer constancy condition'. They also refer to this condition as 'Geometric Conservation Laws', but no reference is given for either denomination. However, since the seminal paper

   Lin, S. J., Rood, R. B. (1996). Multidimensional flux form semi-Lagrangian transport schemes. Monthly Weather Review, 124(9), 2046-2070

   it has become customary to describe this condition as 'consistency with continuity' or 'compatibility with continuity', see e.g.

Gross, E. S., Bonaventura, L., Rosatti, G. (2002). Consistency with continuity in conservative advection schemes for freesurface models. International Journal for Numerical Methods in Fluids, 38(4), 307-327.

Fringer, O. B., Gerritsen, M., Street, R. L. (2006). An unstructured-grid, finite-volume, nonhydrostatic, parallel coastal ocean simulator. Ocean modelling, 14(3-4), 139-173.

Kuhnlein, C., Smolarkiewicz, P. K., Dornbrack, A. (2012). Modelling atmospheric flows with adaptive moving meshes. Journal of Computational Physics, 231(7), 2741-2763.

The authors might consider using (also) this terminology in the revised version. More importantly, as shown in (Gross et al. 2002), this consistency/compatibility must be guaranteed also for the time discretizations of the tracer and continuity equations. Apparently, this aspect is not discussed in the preprint, so that the consistency proof provided by the authors cannot be considered complete. This is especially important for semi-implicit discretizations, since using advecting velocities at different time levels might easily occur in this context. Completing the discussion on this aspect would definitely increase the value of the preprint. Furthermore, in the numerical experiments this property is only checked in a case with flat bottom, while a numerical check also for more complex bathymetry is necessary. A mandatory revision is to include similar checks of the preservation of constants also for the sloping channel and Venice lagoon benchmarks.

3) In many parts of the paper, the authors try to consider different $z-$ coordinate formulations within the same multi-layer framework. While this is definitely a positive thing to do and a potentially important contribution of the preprint, often the way in which the different formulations are handled is confusing, also because of the related lack of specific definition of the $l_\alpha$ coefficients. The authors are strongly suggested to review all the parts of the text in which the different formulations are presented and make sure that all the quantities involved are properly defined and the specific steps to be taken for each formulation are described completely and in detail.

4) In the introduction (line 40) the authors claim that their remeshing strategy solves possible stability problems of approaches proposed earlier in the literature. This claim is repeated later in the preprint (page 9, line 202). However, no stability analysis is provided to support this claim. In the revised version, the authors should either provide a proof of stability for the proposed algorithm or remove/reformulate any claims of superior stability properties.

5) At the end of section 3.2 (line 175) the authors introduce a pseudo-time quantity $\tau$ which is then discretized in steps $\Delta\tau$ to proceed to the remapping of discrete quantities, see equation (15). However, there is no indication on how this pseudo-time step should be chosen and on whether any empirical or theoretical bounds should be respected to maintain stability. Inclusion of some criterion for the choice of $\Delta\tau$ (sufficiently small fraction of $\Delta t$?) is mandatory for the revised version.

6) The truncation error analysis presented in the appendix uses in an essential way the linearized equation

$$\partial_t\zeta + H_0\partial_x u = 0.$$

This form is consistent with the assumption that the linearization has been performed around the constant state $U = 0, H = H_0$ and that the velocity field $u$ is a first order perturbation. This seems however inconsistent with the assumption of an $O(1)$ tidal amplitude $A$. The correct linearized equation would be in this case

$$\partial_t\zeta + U\partial_x\zeta + H_0\partial_x u = 0$$

with $U = O(1)$. As a consequence, the upper bound on the divergence would also depend on the free surface gradients, which would seem physically reasonable. The whole derivation in the appendix would have to be reformulated taking into account the correct linearized equation. Furthermore, the following numerical experiments seem to consider a constant laminar diffusivity, which is rather different from the turbulent profiles that would typically arise in a realistic situation, making the whole discussion somewhat academic. Either the authors find a way to address this major shortcomings of the analysis presented in the appendix, or they would be strongly suggested to remove this analysis which is only marginally related to the main topic of the paper.

7) The model equations are written in dimensional form, so measure units should be introduced for all the quantities reported when describing the numerical experiments (they are missing in section 5.1)

**Technical corrections:**

1) line 13: replace 'that follow the materials' with 'that are material surfaces'

2) line 21: the reference to (Cheng et al. 1993) could be complemented with a reference to the unstructured UNTRIM-3D model, such as e.g.

   Casulli, V., Walters, R. A. (2000). An unstructured grid, threedimensional model based on the shallow water equations. International journal for numerical methods in fluids, 32(3), 331-348.

   Casulli, V., Zanolli, P. (2002). Semi-implicit numerical modeling of nonhydrostatic free-surface flows for environmental problems. Mathematical and computer modelling, 36(9-10), 1131-1149.

3) line 87: the explicit definition of the term $IPG_\alpha$ should be introduced

4) line 107: in formula (8), the argument of the flux limiter should be specified; this should also be done in all the other points where this quantity is introduced, most of the time without specification of the argument and of the location at which it is computed; also in formula (9), a $\phi_\alpha$ appears that is not defined anywhere

5) line 114: in formula (9), the second derivatives on the right hand side have no reason to be positive, so that this kind of upper bound should be performed considering only absolute values of the quantities involved

6) line 135-139: what the authors denote as the space discrete and fully discrete variables, respectively, are indeed (see e.g. formula (17)) the $P^1$ finite element approximation of the solution, which is a piecewise polynomial continuous function; the text should be changed to avoid this confusion; even though the notation $u_h(x)$ is customary in the finite element literature, it is a bit confusing in a context where $h$ has a different meaning (the finite element 'h' would correspond to what in

the preprint is called $\Delta x_E$.)

7) line 146: the first sentence of section 3.1 is superfluous, any time discretization method will update the free surface based on equation (3)...the sentence should either be removed or reformulated if something else was meant

8) line 158: coefficients $l_{\alpha,i}$ are introduced without having been previously defined; this is related to point 1) in the specific comments above; clear definition of these quantities is essential, since otherwise the proposed methods are not completely defined nor reproducible

9) line 188: it is unclear what do the authors mean by 'z-layer depth at rest $\Delta z_\alpha^0$; if this is the depth at the initial time, it is better to say so because what 'at rest' means in a hydrodynamical simulation is very unclear

10) line 217: the expression 'hanging interfaces' is probably derived from the 'hanging nodes' used in the literature on numerical methods for non conformal meshes; however, while hanging nodes makes sense (the quadrature nodes on one side do not have a counterpart on the other side and numerical fluxes or mortar procedures must be employed), hanging interfaces does not make much sense in my opinion, since the interface is a perfectly well defined geometrical object; the authors are strongly suggested to modify this terminology and use instead e.g. non-conformal boxes, as they do in the following

11) line 221: change 'sophisticate' into 'complicate'

12) line 240: change 'kernel' into 'basis function'

13) line 244: the formula below this line is obtained according to the authors by integration by parts, but contains no boundary terms, the authors should explain whether these terms are zero and why or correct the formula

14) line 286: the rule used to define the mesh size should be explicitly reported, e.g. $h_K$ as the maximum length of the triangle sides

15) in the caption of Figure 8, the quantities $T, T_0$ used to define the relative tracer conservation error are not defined; if they are meant to be the total tracer mass at the end and at the beginning of the simulation, then $|T - T_0|$ (absolute value is missing in the text!) is the absolute

error, not the relative error; the authors are suggested to display values of $|T - T_0|/|T_0|$

16) line 333: replace 'summerized' with 'summarized'

17) line 437: replace 'its' with 'her'!!! it would also be appropriate to specify better the direct or indirect contribution of Dr. Bellafiore to this work

18) general comment: personally I think it is graphically better to write $z-$coordinate than z-coordinate; this is not a required change but I think that it would be appropriate

---

## Referee Comment (RC3)

[referee-annotated manuscript omitted]

---

## Author Comment (AC1)

**Reply to Reviewer 1**

Dear Prof. Bonaventura,

Thank you for considering the idea of the manuscript relevant for the scope of the journal. We are very grateful for all your remarks. We will try to answer to the individual points raised in a separate document that follows.

Sincerely,

The authors,
Luca Arpaia, C. Ferrarin, M. Bajo and G. Umgiesser

**Specific comments**

1) Starting with equations 3-4 the authors introduce an undefined quantity $h$ that seems to play the role of vertical spacing. This quantity appears alongside the quantity $h_\alpha$ , which instead is defined on page 4. Later on, quantities $l_\alpha$ are introduced and employed without being defined. Whether these have the same meaning usually implied in the multilayer literature is never stated. This does not allow the reader to understand exactly how the proposed numerical method is formulated and makes the results described in the preprint impossible to reproduce. A mandatory revision is to define properly all quantities that are being introduced and explain clearly their relationship (if any) to analogous quantities introduced in the literature on multi-layer models. Furtermore, in the appendix (line 386) the authors seem to imply that in models for which $h_\alpha = l_\alpha H$ the mass transfer terms across the layers have to be zero. However, this is not true for the (Audusse et al. 2011) paper which apparently constitutes the reference for the present paper nor for the related discretization approach introduced in Fernandez-Nieto, E. D., Kone, E. H., Chacon Rebollo, T. (2014). A multilayer method for the hydrostatic Navier-Stokes equations: a particular weak solution. Journal of Scientific Computing, 60, 408-437. and employed in (Bonaventura et al. 2018). The authors should clearly specify to which multi-layer formula.

*As you remarked, we intended $hu_\alpha = h_\alpha u_\alpha$ with $u_\alpha$ the average value defined in (2). We will change such unclear notation to the standard one. Moreover the total water depth $H = \zeta + b$ has never been defined and this may have generated confusion. In the appendix we have analysed a very idealised case: a barotropic tide with no bottom friction ($u_\alpha(x,t) = u(x,t)$ $\alpha = 1, ..., N$) over flat bathymetry. In this case the mass-transfer at the layer interfaces for the $\sigma-coordinate$ case (or $z-star$) collapses to the depth integrated mass conservation $G_{\alpha-1/2} = (\partial_t H + \partial_x (Hu))\sum_{\beta=N}^{\alpha} l_\beta = 0$, and it is thus zero. That sentence was not referred to the general case where the mass-transfer across the layers is not zero. As you suggested, the appendix will be removed.*

2) The authors devote a significant effort to the important issue of proving that what they call the 'tracer constancy condition'. They also refer to this condition as 'Geometric Conservation Laws', but no reference is given for either denomination. However, since the seminal paper Lin, S. J., Rood, R. B. (1996). Multidimensional flux form semi- Lagrangian transport schemes. Monthly Weather Review, 124(9), 2046-2070 it has become customary to describe this condition as 'consistency with continuity' or 'compatibility with continuity', see e.g.Gross, E. S., Bonaventura, L., Rosatti, G. (2002). Consistency with continuity in conservative advection schemes for freesurface models. In- ternational Journal for Numerical Methods in Fluids, 38(4), 307-327. Fringer, O. B., Gerritsen, M., Street, R. L. (2006). An unstructured-grid, finite-volume, nonhydrostatic, parallel coastal ocean simulator. Ocean modelling, 14(3-4), 139-173. Kuhnlein, C., Smolarkiewicz, P. K., Dornbrack, A. (2012). Modelling atmospheric flows with adaptive moving meshes. Journal of Computa- tional Physics, 231(7), 2741-2763. The authors might consider using (also) this terminology in the revised version. More importantly, as shown in (Gross et al. 2002), this consistency/compatibility must be guaranteed also for the time discretiza- tions of the tracer and continuity equations. Apparently, this aspect is not discussed in the preprint, so that the consistency proof provided by the authors cannot be considered complete. This is especially important for semi-implicit discretizations, since using advecting velocities at different time levels might easily occur in this context. Completing the discussion on this aspect would definitely increase the value of the preprint. Furthermore, in the numerical experiments this property is only checked in a case with flat bottom, while a numerical check also for more complex bathymetry is necessary. A mandatory revision is to include similar checks of the preservation of constants also for the sloping channel and Venice lagoon benchmarks.

*Thank you for the references. We will improve the bibliography, adding a paragraph in the introduction with the references. We will switch the condition name too, taking out the ambiguous sentence on the GCL.*

*Concerning the tracer constancy, in the manuscript we have not considered the time discretization and we focus only on how the variable number of layers may impact the tracer constancy. In SHYFEM the*

*time-discrete layerwise mass-equation reads:*

$$G_{\alpha-1/2}^{n+1} = G_{\alpha+1/2}^{n+1} + \frac{h_\alpha^{n+1} - h_\alpha^n}{\Delta t} + \frac{\partial}{\partial x} (h_\alpha u_\alpha)^{n+1/2} \tag{1}$$

*Then the tracer is updated with: 1/ a θ-method 2/ to answer your question, horizontal transport is computed at $n + 1/2$, while the vertical mass-transfer function is at $n + 1$:*

$$\frac{(h_\alpha t_\alpha)^{n+1} - (h_\alpha t_\alpha)^n}{\Delta t} + \frac{\partial}{\partial x} \left( (h_\alpha u_\alpha)^{n+1/2} t_\alpha^{n+\theta_h} \right) = [G^{n+1} t^{n+\theta_v}]_{\alpha+1/2}^{\alpha-1/2} \tag{2}$$

*We agree that choice 2/ is also important to verify tracer constancy. We will correct/remove "Assuming that the time derivative and the vertical advection terms in (6) and (7) are treated equally, it is enough to verify that the horizontal advection term reduces to the mass-flux term". We will add the time discretization in a proper form, considering also the effect of a variable number of layers and of remaps. Remaps do not destroy the constancy property.*

*Concerning more complex cases, especially wetting/drying can be tricky. Although, since (1) and (2) are still valid, we do expect to conserve mass and preserve tracer constancy at wet/dry and dry nodes; in the revised manuscript we will show the verification (or not) for all the tests.*

3) In many parts of the paper, the authors try to consider different $z-$ coordinate formulations within the same multi-layer framework. While this is definitely a positive thing to do and a potentially important contribution of the preprint, often the way in which the different formulations are handled is confusing, also because of the related lack of specific definition of the $l_\alpha$ coefficients. The authors are strongly suggested to review all the parts of the text in which the different formulations are presented and make sure that all the quantities involved are properly defined and the specific steps to be taken for each formulation are described completely and in detail.

*We agree with you that the clarity may be improved by defining properly the interface position $z_{\alpha+1/2}$ the layer thickness $h_\alpha$ for the different vertical coordinates systems. We realized that all these definitions are either at continuous level (like z in (1)) or spread out over different sections. We will rewrite Section 2 in a more structured fashion. First we will present the layerwise Shallow Water model. Then, in the same section, we will close the problem defining the evolution of the interfaces $z_{\alpha+1/2}$ and of $h_\alpha$, for the different z-systems introduced.*

*We consider a transformation from a reference domain $x \in [0, L]$, $z \in [0, -b(x)]$ discretized vertically with flat interfaces $Z_{1/2} = 0, Z_{1+1/2}, ... Z_{\alpha+1/2}, ... Z_{N+1/2} = -\max b(x)$ to a physical domain $x \in [0, L]$, $z \in [\zeta(x,t), -b(x)]$ with interfaces $z_{1/2} = \zeta(x,t), z_{1+1/2}, ... z_{\alpha+1/2}(x,t), ... z_{N+1/2} = -\max b(x)$. For $z-$star, the transformation at a discrete level reads:*

$$z_{\alpha+1/2}(x,t) = \zeta(x,t) + S_{\alpha+1/2}(x) \left( \zeta(x,t) + b(x) \right)$$

*with $S_{\alpha+1/2}(x) = \frac{Z_{\alpha+1/2}}{b(x)}$. The layer thickness can be deduced from the total water depth:*

$$h_\alpha(x,t) = z_{\alpha-1/2} - z_{\alpha+1/2} = l_\alpha(x) \left( \zeta(x,t) + b(x) \right) = l_\alpha(x) H(x,t)$$

*with $l_\alpha(x) = \frac{Z_{\alpha-1/2} - Z_{\alpha+1/2}}{b(x)} = \frac{\Delta Z_\alpha}{b(x)}$ which is prescribed by the reference grid and satisfy $\sum_{\alpha=1}^N l_\alpha(x) = 1$. Except for the fact that $l_\alpha = l_\alpha(x)$, we believe it has the analogous meaning as in the multi-layer literature.*

4) In the introduction (line 40) the authors claim that their remeshing strategy solves possible stability problems of approaches proposed earlier in the literature. This claim is repeated later in the preprint (page 9, line 202). However, no stability analysis is provided to support this claim. In the revised version, the authors should either provide a proof of stability for the proposed algorithm or remove/reformulate any claims of superior stability properties.

*The sentence will be removed and we can clarify what we wanted to point out. Our approach consists in considering the area swept by the interface as the sum of two contributions: one due to the grid movement with velocity $\sigma_{mov}$ and one due to the collapse of the element with grid velocity $\sigma_{top}$, see fig.2 (in the manuscript unfortunately both the interface velocities have the identical symbol $\sigma$). Without such an interpretation, one may be tempted for example to perform a removal of a surface layer after the semi-implicit update (no step 3.1, only step 3.2 of the proposed algorithm). Then a surface layer with negative thickness can occur in the semi-implicit update, unless the timestep is somehow limited a-posteriori with an iterative procedure to avoid the appereence of negative layer thickness. While a linear stability analysis would be interesting it seems quite complex (there is a non-linearity in the algorithm because the grid is deformed or not, depending on the free-surface position). We would prefer to focus on the clarification of other points.*

5) At the end of section 3.2 (line 175) the authors introduce a pseudo-time quantity $\tau$ which is then discretized in steps $\Delta\tau$ to proceed to the remapping of discrete quantities, see equation (15). However, there is no indication on how this pseudo-time step should be chosen and on whether any empirical or theoretical bounds should be respected to maintain stability. Inclusion of some criterion for the choice of $\Delta\tau$ (sufficiently small fraction of $\Delta t$?) is mandatory for the revised version.

*The pseudotime is $\tau = (t - t^n)$. It is introduced because, instead of solving (4),(7) at once with $\sigma = \sigma_{mov} + \sigma_{top}$, we have considered a splitting procedure. First we solve the governing equations on a moving grid, that is the discrete counterpart of eq. (4) or (7) with $\sigma_{mov}$. Then we solve equation (15) with $\sigma_{top}$. In the implementation of eq. (15) we made the simplest choice: set a grid velocity that conserves the volume ($\sigma_{top} = \Delta z / \Delta t$), upwind flux and Explicit Euler. In the case of layer 1 removal (fig.2 "layer collapse" of the manuscript) we have the interface $2 - 1/2$ which goes towards $1 - 1/2$, thus moving upward; eq. (15) for the layer 2 reduces to (we neglect superscript $n + 1$):*

$$\begin{aligned}
\widetilde{h_2 u_2} &= h_2 u_2 + \Delta\tau [\sigma_{top} u]^{2-1/2}_{2+1/2} \\
&= h_2 u_2 + \Delta\tau \left( \frac{\Delta z_{2-1/2}}{\Delta t} u_{2-1/2} \right) \\
&= h_2 u_2 + \frac{\Delta\tau}{\Delta t} h_1 u_1
\end{aligned}$$

*with $\Delta\tau = \Delta t$, $\widetilde{u_2} = (h_2/\widetilde{h_2}) u_2 + (h_1/\widetilde{h_2}) u_1$. Since $\widetilde{h_2} = h_1 + h_2$ we have that $\min(u_1, u_2) \le \widetilde{u_2} \le \max(u_1, u_2)$. Sorry for the lack of clarity, we will correct this part.*

6) The truncation error analysis presented in the appendix uses in an essential way the linearized equation

$$\partial_t \zeta + H_0 \partial_x u = 0$$

This form is consistent with the assumption that the linearization has been performed around the constant state $U = 0$, $H = H_0$ and that the velocity field u is a first order perturbation. This seems however inconsistent with the assumption of an $O(1)$ tidal amplitude A. The correct linearized equation would be in this case

$$\partial_t \zeta + U \partial_x \zeta + H_0 \partial_x u = 0$$

with $U = O(1)$. As a consequence, the upper bound on the divergence would also depend on the free surface gradients, which would seem physically reasonable. The whole derivation in the appendix would have to be reformulated taking into account the correct linearized equation. Furthermore, the following numerical experiments seem to consider a constant laminar diffusivity, which is rather different from the turbulent profiles that would typically arise in a realistic situation, making the whole discussion somewhat academic. Either the authors find a way to address this major shortcomings of the analysis presented in the appendix, or they would be strongly suggested to remove this analysis which is only marginally related to the main topic of the paper.

*We will remove the appedix with the analysis. We wanted to understand the differences between the $z-$star and the $z$ simulations in coastal environments with stratification and tide. We agree that 1) it is not relevant to the implementation with insertion and removal of layers 2) being too simplified at the end it was not really helpful to guide us in the interpratation of the results. Just for our knowledge, the tidal amplitude is $O(1)$ but the depth is still $H_0 = 50\,m$, $\epsilon = A/H_0 \leq 0.05$. Probably was this value too high for the linearization to be valid? Thank you.*

7) The model equations are written in dimensional form, so measure units should be introduced for all the quantities reported when describing the numerical experiments (they are missing in section 5.1)

*Yes, thank you. We will add them.*

**Technical corrections**

1) line 13: replace 'that follow the materials' with 'that are material surfaces

2) line 21: the reference to (Cheng et al. 1993) could be complemented with a reference to the unstructured UNTRIM-3D model, such as e.g. Casulli, V., Walters, R. A. (2000). An unstructured grid, threedimensional model based on the shallow water equations. International jour- nal for numerical methods in fluids, 32(3), 331-348.

11) line 221: change 'sophisticate' into 'complicate'

12) line 240: change 'kernel' into 'basis function'

16) line 333: replace 'summerized' with 'summarized'

18) general comment: personally I think it is graphically better to write $z-$ coordinate than z-coordinate; this is not a required change but I think that it would be appropriate

*Thank you we will correct the typos, sentences and the references.*

3) line 87: the explicit definition of the term $IPG_\alpha$ should be introduced

*For the internal pressure term SHYFEM use the density Jacobian form:*

$$IPG_\alpha = h_\alpha g b(\zeta)\frac{\partial\zeta}{\partial x} + g\int_{z_{\alpha+1/2}}^{z_{\alpha-1/2}}\int_{z}^{\zeta} J(b,z')dz'\,dz$$

*with $J(b,z) = \frac{\partial b}{\partial x}\big|_s - \frac{\partial b}{\partial z}\,\frac{\partial z}{\partial x}\big|_s$ the density Jacobian ($b = \frac{\rho_0-\rho}{\rho_0}$ the buoyancy). The integral is performed by interface:*

$$\int_{z_{\alpha+1/2}}^{z_{\alpha-1/2}}\int_{z_\alpha}^{\zeta} J(b,z')dz' = h_\alpha\sum_{\beta=1}^{\alpha} J(b_{\beta-1/2}, z_{\beta-1/2})h_{\beta-1/2}$$

*which means evaluating the Jacobian at the interface location, with a standard formula that can be found in [Shchepetkin and McWilliams, Journal of Geophysical Research, 2003 formula 2.3] or [Klingbeil et al., Ocean Modelling, 2018 formula 7.5]. We will add these formulas with the details in the text.*

4) line 107: in formula (8), the argument of the flux limiter should be specified; this should also be done in all the other points where this quantity is introduced, most of the time without specification of the argument and of the location at which it is computed; also in formula (9), a $\phi_\alpha$ appears that is not defined anywhere

*We will specify both the argument and the location of the limiter $\phi(r_{\alpha-1/2})$. Formula (9) will disappear with the appendix.*

5) line 114: in formula (9), the second derivatives on the right hand side have no reason to be positive, so that this kind of upper bound should be performed considering only absolute values of the quantities involved

*Thank you for the correction, it is a mistake. We will remove the appendix and formula (9) with it, see point 6 of Specific comments.*

6) line 135-139: what the authors denote as the space discrete and fully discrete variables, respectively, are indeed (see e.g. formula (17)) the $P^1$ finite element approximation of the solution, which is a piecewise polynomial continuous function; the text should be changed to avoid this confusion; even though the notation $u_h(x)$ is customary in the finite element literature, it is a bit confusing in a context where h has a different meaning (the finite element 'h' would correspond to what in the preprint is called $\Delta x_E$.)

*We think the confusion comes from the fact that we wanted to describe the $z-$ surface-adaptive coordinate regardless of the horizontal spatial discretization (finite volume, finite element). And then some definitions conflict with the finite element notation introduced for the tracer constancy verification. We can discuss only the staggered finite element case. To avoid $(\cdot)_h$ we can change it to $(\cdot)_{\Delta x}$ or we can omit the subscript, with an abuse of notation. e.g. for formula (18) "We seek an approximation, still denoted by $\zeta$ with an abuse of notation, which belongs to the finite dimensional space .... ". We will think how to clarify the notation.*

7) line 146: the first sentence of section 3.1 is superfluous, any time discretization method will update the free surface based on equation (3)...the sentence should either be removed or reformulated if something else was meant

*We will remove the sentence.*

8) line 158: coefficients $l_{\alpha,i}$ are introduced without having been previously defined; this is related to point 1) in the specific comments above; clear definition of these quantities is essential, since otherwise the proposed methods are not completely defined nor reproducible

*See answer to Specific comment 3)*

9) line 188: it is unclear what do the authors mean by 'z-layer depth at rest $\Delta z_\alpha^0$ ; if this is the depth at the initial time, it is better to say so because what 'at rest' means in a hydrodynamical simulation is very unclear

*With "at rest" we meant the reference configuration (see above, specific comment 3) which for us is always the one with $\zeta(x) = 0$ ("at rest"). Actually, it's true that the notation with $\Delta z_\alpha^0$ is not clear because it seems to suggest the initial time. In the revised manuscript we will use instead capital letters for the reference configuration $\to \Delta Z_\alpha$ (see always specific comment 3).*

10) line 217: the expression 'hanging interfaces' is probably derived from the 'hanging nodes' used in the literature on numerical methods for non conformal meshes; however, while hanging nodes makes sense (the quadrature nodes on one side do not have a counterpart on the other side and numerical fluxes or mortar procedures must be employed), hanging interfaces does not make much sense in my opinion, since the interface is a perfectly well defined geometrical object; the authors are strongly suggested to modify this terminology and use instead e.g. non-conformal boxes, as they do in the following

*We agree that, in the literature, "hanging interface" never appears. We will modify this expression.*

13) line 244: the formula below this line is obtained according to the authors by integration by parts, but contains no boundary terms, the authors should explain whether these terms are zero and why or correct the formula

*The boundary term has been neglected. We will add a comment on boundary conditions.*

14) line 286: the rule used to define the mesh size should be explicitly reported, e.g. $h_K$ as the maximum length of the triangle sides

*We will be more precise about the mesh size.*

16) in the caption of Figure 8, the quantities $T$, $T_0$ used to define the relative tracer conservation error are not defined; if they are meant to be the total tracer mass at the end and at the beginning of the simulation, then $|T - T_0|$ (absolute value is missing in the text!) is the absolute error, not the relative error; the authors are suggested to display values of $|T - T_0|/|T_0|$

*We confirm that the values shown are not relative but absolute. There is an error in the caption of the figure. We will show the relative errors with a modified caption.*

17) line 437: replace 'its' with 'her' !!! it would also be appropriate to specify better the direct or indirect contribution of Dr. Bellafiore to this work

*We are really sorry, this is a bad mistake due to english deficiency*

---

## Author Comment (AC3)

**Reply to Reviewer 2**

Dear Dr. Capodaglio,

Thank you for considering the idea of the manuscript relevant for the scope of the journal. We are grateful for your feedback and suggestions. We aim to enhance the presentation's clarity and rigor while incorporating additional tests. We address the individual points raised in a separate document that follows.

Sincerely,

The authors,
Luca Arpaia, C. Ferrarin, M. Bajo and G. Umgiesser

**General comments**

1) The paper is currently suffering from a lack of mathematical rigor and the presentation is sometimes confused. The notation is often unnecessarily complex and some definitions are missing. Please refer to the comments in the attached PDF document and revise the presentation (especially in Section 2).

*In the revised version, Section 2 will be restructured. First we will present the multi-layer (or layerwise) shallow water model, eq.(3),(4) and (7) with improved references and notation. Then, in the same section, we will close the problem defining the generalized vertical transformation (1) in a discrete form which determines the evolution of the interfaces $z_{\alpha+1/2}$ for the different z-coordinates introduced (z and z−star).*

*The main issues in Section 3 are the remap part and paragraph 3.3: both need a clarification. Concerning the first point, our approach consists in considering the area swept by the interface in one time-step as the sum of two contributions: one due to the grid movement with velocity $\sigma_{mov}$ and one due to the collapse of the element with grid velocity $\sigma_{top}$, see fig.2 (in the manuscript unfortunately both the interface velocities have the identical symbol $\sigma$, this will be changed). The pseudotime is $\tau = (t − t^n)$. It is introduced because, instead of solving (4),(7) at once with $\sigma = \sigma_{mov} + \sigma_{top}$, we have considered a splitting procedure. First we solve the governing equations on a moving grid, that is the discrete counterpart of eq. (4) or (7) with $\sigma_{mov}$. Then we solve equation (15) with $\sigma_{top}$. We will review this part. We will also consider the comments provided in Section 3.3.*

*Concerning your main remarks in the pdf, related to these aspects:*

- *Line 65. We will add a sketch for the multilayer notation, similar to the one in the figure attached (which is the same as [E. Audusse et al. A Multilayer Saint-Venant system with mass exchanges for shallow water flows. Derivation and numerical validation. ESAIM, 2011] but with a reversed layer ordering).*

- *Line 65,66,70,71,75. This part could be rewritten in a vertical discrete framework as follow ($\zeta(x,t)$ is the free-surface, $b(x)$ the bathymetry, see again the figure):*
  *"Consider a transformation from a reference domain $x \in [0,L]$, $z \in [0, −b(x)]$ discretized vertically with flat interfaces $Z_{1/2} = 0, Z_{1+1/2}, ...Z_{\alpha+1/2}, ...Z_{N−1/2}$ to a physical domain $x \in [0,L]$, $z \in [\zeta(x,t), −b(x)]$ with interfaces $z_{1/2} = \zeta(x,t), z_{1+1/2}, ...z_{\alpha+1/2}(x,t), ...z_{N−1/2}$. For z−star, the transformation at a discrete level reads:*

  $$z_{\alpha+1/2}(x,t) = \zeta(x,t) + S_{\alpha+1/2}(x)\left(\zeta(x,t) + b(x)\right)$$

  *with $S_{\alpha+1/2}(x) = \frac{Z_{\alpha+1/2}}{b(x)}$. For standard z−coordinate we can use the following definition for the internal interfaces:*

  $$z_{\alpha+1/2} = Z_{\alpha+1/2}$$

  *".*

- *missing definitions may help clarify the text and will be added (e.g. square brackets for jumps $[\cdot]$ formula (4), finite element notation $(\cdot)_h$ line 136, simplification of the symbols in formulas line 155, 157).*

- *the notation will be changed when it is confusing (e.g. capital letter for tracers, notation of internal pressure gradient) or not rigorous (line 170).*

- *proper references will be added. For example, the name tracer constancy preservation (line 98 and 227) has been taken from [Shchepetkin and McWilliams, Ocean Modelling, 2005].*

- *we can rephrase unclear word/sentences, being more precise: e.g. Line 150 from "the layers spanned by the free-surface" to "the layers whose top-interface crosses or is above the free-surface", Line 235 from "form" to "basis", Line 240 "kernel" to "basis" etc ...*

[Figure]

*Figure. Sketch of the reference (left) and physical (right) space*

2) The authors are doing a good job in the numerical results with showing that the proposed method is well behaved and that it does not perform worse than z-star. Although it is my opinion that additional results are missing to show that the extra hassle of introducing or removing layers is actually beneficial for some application. The authors should include at least one additional test case to show that their method enables simulations that are currently not possible with other existing z-type coordinate models. If this is deemed unfeasible, please show at least that there exists one relevant test case for which your model clearly outperforms existing z-coordinate models.

*The reviewer raised a good point. We agree with him that this part is missing in the manuscript. We would have expected some performance differences between $z-$surface-adaptive and $z-$star coordinate, in intertidal flats computations. This is because the vertical discretization of the water column differs significantly in the two cases (with $z-$surface-adaptive one mainly performs runup/rundown with a one-layer shallow water model, with z-star the initial number of layers remains constant during the simulation). For the test #2 (tidal channel) such performance differences are still not very clear to us. We intend to further assess the two methods by evaluating their performance on additional wetting/drying benchmarks. Finally, in light of the limited differences observed in the Venice Lagoon test case, we plan to substitute it with the SHYFEM application to the Po Delta where more significant disparities between the two methods can be observed in reproducing salinity stratification and salt water intrusion.*

3) The computational performance of the method is relegated to the last 4 lines of the numerical results section. Parallel performance is clearly of paramount interest for your readers so a more detailed analysis should be included in the paper. Currently, in the serial runs reported, the overhead of the present method is 8% over z-star. Would that get worse in parallel? If so, what are possible avenues for mitigation?

*We can add computational performances for all the tests. We have not covered parallel implementation aspects because all the tests have been accomplished with a serial run. Although a recent version of SHYFEM is parallelized with MPI, the branch with the $z-$surface-adaptive developments only supports a partial parallelization with OMP. Unfortunately, this means that we cannot evaluate parallel performances of the $z-$surface-adaptive algorithm in the short term. We can comment that the algorithm (grid movement, insertion/removal) mainly operates on the vertical grid, and the parallel execution of these tasks should not encounter any issues. The stencil of the numerical scheme is not enlarged with respect to the standard method. However some variables have been introduced only for the insertion/removal operations. This is the case of the nodal top layer index (10) which must be exchanged between the domains.*

---

## Author Response (AR1)

Dear Prof. Bonaventura and Dr. Capodaglio,

Thank you for considering the idea of the manuscript relevant for the scope of the journal. We are very grateful for all your remarks and feedback that were all very helpful. In summary, the key modifications are as follows:

- The main change was devoted to improve the clarity and the rigor of the Section 2, the one describing the multilayer model with $z$-layers. In the revised manuscript we started introducing the standard terrain/surface following multilayer model. Then we have tried to give the expression of the layers thickness for $z-$star layers.

- The second change concerns the discretization part, Section 3. Instead of keeping the discussion general, we have focused only on our implementation: semi-implicit staggered finite element. We have provided the discrete equations with a certain level of detail. We have provided a complete proof of the consistency of the discrete tracer equation with the continuity.

- Third, we have revised the discussion on the $z$-layers with layer insertion/removal. We think to have enhanced the clarity of Section 4.

- Fourth, we have substituted the Venice Lagoon test with an idealized test of the Po Delta, where the $z$-layers and $z$-star exhibited noticeable differences.

We have tried to answer to the individual points raised in a separate document that follows.

Sincerely,

The authors,
Luca Arpaia, C. Ferrarin, M. Bajo and G. Umgiesser

**Reply to Reviewer 1**

**Specific comments**

1) Starting with equations 3-4 the authors introduce an undefined quantity $h$ that seems to play the role of vertical spacing. This quantity appears alongside the quantity $h_\alpha$ , which instead is defined on page 4. Later on, quantities $l_\alpha$ are introduced and employed without being defined. Whether these have the same meaning usually implied in the multilayer literature is never stated. This does not allow the reader to understand exactly how the proposed numerical method is formulated and makes the results described in the preprint impossible to reproduce. A mandatory revision is to define properly all quantities that are being introduced and explain clearly their relationship (if any) to analogous quantities introduced in the literature on multi-layer models. Furtermore, in the appendix (line 386) the authors seem to imply that in models for which $h_\alpha = l_\alpha H$ the mass transfer terms across the layers have to be zero. However, this is not true for the (Audusse et al. 2011) paper which apparently constitutes the reference for the present paper nor for the related discretization approach introduced in Fernandez-Nieto, E. D., Kone, E. H., Chacon Rebollo, T. (2014). A multilayer method for the hydrostatic Navier-Stokes equations: a particular weak solution. Journal of Scientific Computing, 60, 408-437. and employed in (Bonaventura et al. 2018). The authors should clearly specify to which multi-layer formula.

*As you remarked, we intended $hu_\alpha = h_\alpha u_\alpha$ with $u_\alpha$ the average value. In the revised manuscript we have changed such unclear notation to the standard one. Moreover the total water depth $H = \zeta + z_b$ has never been defined and this may have generated confusion. In the revised manuscript we have strongly restructured Section 2. We have presented the multilayer model for stratified fluid in a form very similar to the one presented in (Audusse et al.: Approximation of the hydrostatic Navier-Stokes system for density stratified flows by a multilayer model: Kinetic interpretation and numerical solution, J. Comput. Phys., 230, 3453–3478, 2011). They differ for the more stringent Boussinesq assumption used in our study and for the expression of the pressure gradient term, written with a pressure Jacobian form in the reference while, in our case, it has been further manipulated. It has been split into an external and an internal pressure gradient. The latter is then written in the density Jacobian (DJ) form. Both the Boussinesq assumption and the DJ pressure gradient are standard choices for coastal ocean models.*

*Thank you for the interesting reference [Fernandez-Nieto,2014]. In the revised manuscript we have given both the definition of the mass-transfer used in practice (eq. 14 or 15) and also the formal definition of the mass-transfer function. The latter allows us to highlight the contributions of the vertical velocity and more important of the vertical grid velocity, which we use later in our method. For the formal definition we have used the one from the derivation of [Fernandez-Nieto,2014] that, as we have understood, is compatible with the eq. (14).*

*In the appendix we have analysed a very idealised case: a barotropic tide with no bottom friction ($u_\alpha(x,t) = u(x,t)$ $\alpha = 1,...,N$) over flat bathymetry. In this case the mass-transfer at the layer interfaces for the $\sigma-$coordinate case (or $z-$star) collapses to the depth integrated mass conservation $G_{\alpha-1/2} = (\partial_t H + \partial_x(Hu))\sum_{\beta=N}^{\alpha} l_\beta = 0$, and it is thus zero. That sentence was not referred to the general case where the mass-transfer across the layers is not zero. As you suggested, the appendix has been removed.*

2) The authors devote a significant effort to the important issue of proving that what they call the 'tracer constancy condition'. They also refer to this condition as 'Geometric Conservation Laws', but no reference is given for either denomination. However, since the seminal paper

Lin, S. J., Rood, R. B. (1996). Multidimensional flux form semi- Lagrangian transport schemes. Monthly Weather Review, 124(9), 2046-2070

it has become customary to describe this condition as 'consistency with continuity' or 'compatibility with continuity', see

e.g.Gross, E. S., Bonaventura, L., Rosatti, G. (2002). Consistency with continuity in conservative advection schemes for freesurface models. In- ternational Journal for Numerical Methods in Fluids, 38(4), 307-327.

Fringer, O. B., Gerritsen, M., Street, R. L. (2006). An unstructured- grid, finite-volume, nonhydrostatic, parallel coastal ocean simulator. Ocean modelling, 14(3-4), 139-173.

Kuhnlein, C., Smolarkiewicz, P. K., Dornbrack, A. (2012). Modelling atmospheric flows with adaptive moving meshes. Journal of Computa- tional Physics, 231(7), 2741-2763.

The authors might consider using (also) this terminology in the revised version. More importantly, as shown in (Gross et al. 2002), this consistency/compatibility must be guaranteed also for the time discretiza- tions of the tracer and continuity equations. Apparently, this aspect is not discussed in the preprint, so that the consistency proof provided by the authors cannot be considered complete. This is especially important for semi-implicit discretizations, since using advecting velocities at different time levels might easily occur in this context. Completing the discussion on this aspect would definitely increase the value of the preprint. Furthermore, in the numerical experiments this property is only checked in a case with flat bottom, while a numerical check also for more complex bathymetry is necessary. A mandatory revision is to include similar checks of the preservation of constants also for the sloping channel and Venice lagoon benchmarks.

*Thank you for the references. We have improved the bibliography with some of the references. We have also switched the condition name too, taking out the ambiguous sentence on the GCL.*

*In the submitted manuscript we have not considered the time discretization and we have focused only on how the variable number of layers may impact the tracer constancy. In SHYFEM the time-discrete layerwise mass-equation reads:*

$$G_{\alpha-1/2}^{n+1} = G_{\alpha+1/2}^{n+1} + \frac{h_\alpha^{n+1} - h_\alpha^n}{\Delta t} + \frac{\partial}{\partial x} (h_\alpha u_\alpha)^{n+\theta_z} \qquad (1)$$

*Then the tracer is updated with horizontal discharges computed at $n + \theta_z$, and the mass-transfer function at $n + 1$:*

$$\frac{(h_\alpha T_\alpha)^{n+1} - (h_\alpha T_\alpha)^n}{\Delta t} + \frac{\partial}{\partial x} \left( (h_\alpha u_\alpha)^{n+\theta_z} T_\alpha^n \right) = \left[ G^{n+1} T^n \right]_{\alpha+1/2}^{\alpha-1/2} \qquad (2)$$

*We agree that these choices are also important to verify tracer constancy. In the revised manuscript we have added the fully discrete equations, including the tracer equation with these details. Then we have been able to complete the proof for the standard scheme, considering also a node which shares element with a variable number of layers and finally when insertion/removal occur.*

*Concerning more complex cases, especially wetting/drying can be tricky. Wetting and drying has not been detailed but since (1) and (2) are still valid at wet/dry and dry nodes, we are able to conserve mass and preserve tracer constancy; in the revised manuscript we have shown the verification for all the tests.*

3) In many parts of the paper, the authors try to consider different $z-$ coordinate formulations within the same multi-layer framework. While this is definitely a positive thing to do and a potentially important contribution of the preprint, often the way in which the different formulations are handled is confusing, also because of the related lack of specific definition of the $l_\alpha$ coefficients. The authors are strongly suggested to review all the parts of the text in which the different formulations are presented and make sure that all the quantities involved are properly defined and the specific steps to be taken for each formulation are described completely and in detail.

*We agree with you that the clarity may be improved by defining properly the interface position $z_{\alpha+1/2}$ the layer thickness $h_\alpha$ for the different vertical coordinates systems. We realized that all these definitions are either at continuous level (like z in (1)) or spread out over different sections. We have restructured*

*Section 2. First we will have presented the multilayer shallow water model. Then we have closed the problem defining the evolution of the interfaces $z_{\alpha\pm1/2}$ and of the layer thickness $h_\alpha$, for z-star.*

4) In the introduction (line 40) the authors claim that their remeshing strategy solves possible stability problems of approaches proposed earlier in the literature. This claim is repeated later in the preprint (page 9, line 202). However, no stability analysis is provided to support this claim. In the revised version, the authors should either provide a proof of stability for the proposed algorithm or remove/reformulate any claims of superior stability properties.

*In the revised manuscript the sentence has been removed and we have clarified what we wanted to point out. Our approach consists in considering the area swept by the interface as the sum of two contributions: one due to the grid movement with velocity $\sigma_{mov}$ and one due to the collapse of the element with grid velocity $\sigma_{top}$, see fig. 4 (in the manuscript unfortunately both the interface velocities have the identical symbol $\sigma$, this have been changed in the revised version). This assures a positive layer thickness as long as the water depth is positive. Without such an interpretation, one may be tempted for example to perform a removal of a surface layer after the semi-implicit update (without the grid movement step). But a surface layer with negative thickness can occur in the semi-implicit update, unless the timestep is somehow limited a-posteriori with an iterative procedure to avoid the appearance of negative layer thickness. While a linear stability analysis would be interesting it seems quite complex (there is a non-linearity in the algorithm because the grid is deformed or not, depending on the free-surface position). We have preferred to focus on the clarification of the other points.*

5) At the end of section 3.2 (line 175) the authors introduce a pseudo-time quantity $\tau$ which is then discretized in steps $\Delta\tau$ to proceed to the remapping of discrete quantities, see equation (15). However, there is no indication on how this pseudo-time step should be chosen and on whether any empirical or theoretical bounds should be respected to maintain stability. Inclusion of some criterion for the choice of $\Delta\tau$ (sufficiently small fraction of $\Delta t$?) is mandatory for the revised version.

*The pseudotime is $\tau = (t - t^n)$. It was introduced because, instead of solving the governing equations (9,10 of the revised manuscript) at once with $\sigma = \sigma_{mov} + \sigma_{top}$, we have considered a splitting procedure. First we solve the governing equations on a moving grid, that is the discrete counterpart of eq. (9) or (10) with $\sigma_{mov}$. Then we solve equation (38) of the revised manuscript with $\sigma_{top}$. In the implementation of eq. (38) we made the simplest choice: set a grid velocity that conserves the volume ($\sigma_{top} = \Delta z/\Delta t$), upwind flux and Explicit Euler. In the case of layer 1 removal (fig. 4 "layer collapse" of the manuscript) we have the interface $2 - 1/2$ which goes towards $1 - 1/2$, thus moving upward; eq. (38) for the layer 2 reduces to (we neglect superscript $n+1$):*

$$
\begin{aligned}
\widetilde{h_2 u_2} &= h_2 u_2 + \Delta\tau [\sigma_{top} u]_{2+1/2}^{2-1/2} \\
&= h_2 u_2 + \Delta\tau \left( \frac{\Delta z_{2-1/2}}{\Delta t} u_{2-1/2} \right) \\
&= h_2 u_2 + \frac{\Delta\tau}{\Delta t} h_1 u_1
\end{aligned}
$$

*with $\Delta\tau = \Delta t$, $\widetilde{u}_2 = (h_2/\widetilde{h_2})u_2 + (h_1/\widetilde{h_2})u_1$. Since $\widetilde{h_2} = h_1 + h_2$ we have that $\min(u_1, u_2) \leq \widetilde{u}_2 \leq \max(u_1, u_2)$. Sorry for the lack of clarity, we have corrected this part in the revised manuscript. The pseudo time was not necessary and it has been removed.*

6) The truncation error analysis presented in the appendix uses in an essential way the linearized equation

$$
\partial_t \zeta + H_0 \partial_x u = 0
$$

This form is consistent with the assumption that the linearization has been performed around the constant state $U = 0$, $H = H_0$ and that the velocity field u is a first order perturbation. This seems however inconsistent with the assumption of an $O(1)$ tidal amplitude A. The correct linearized equation would be

in this case

$$\partial_t \zeta + U \partial_x \zeta + H_0 \partial_x u = 0$$

with $U = O(1)$. As a consequence, the upper bound on the divergence would also depend on the free surface gradients, which would seem physically reasonable. The whole derivation in the appendix would have to be reformulated taking into account the correct linearized equation. Furthermore, the following numerical experiments seem to consider a constant laminar diffusivity, which is rather different from the turbulent profiles that would typically arise in a realistic situation, making the whole discussion somewhat academic. Either the authors find a way to address this major shortcomings of the analysis presented in the appendix, or they would be strongly suggested to remove this analysis which is only marginally related to the main topic of the paper.

*We have removed the appedix with the analysis. We wanted to understand the differences between the $z-$star and the $z$ simulations in coastal environments with stratification and tide. We agree that 1) it is not relevant to the implementation with insertion and removal of layers 2) being too simplified at the end it was not really helpful to guide us in the interpretation of the results.*

7) The model equations are written in dimensional form, so measure units should be introduced for all the quantities reported when describing the numerical experiments (they are missing in section 5.1)

*Thank you. We have added them in section 6.1 of the revised manuscript.*

**Technical corrections**

1) line 13: replace 'that follow the materials' with 'that are material surfaces

*Done*

2) line 21: the reference to (Cheng et al. 1993) could be complemented with a reference to the unstructured UNTRIM-3D model, such as e.g. Casulli, V., Walters, R. A. (2000). An unstructured grid, threedimensional model based on the shallow water equations. International jour- nal for numerical methods in fluids, 32(3), 331-348.

*Done*

11) line 221: change 'sophisticate' into 'complicate'

*Done*

12) line 240: change 'kernel' into 'basis function'

*Done*

16) line 333: replace 'summerized' with 'summarized'

*Thank you. Done*

18) general comment: personally I think it is graphically better to write $z-$ coordinate than z-coordinate; this is not a required change but I think that it would be appropriate

*Thank you for this advice; it is greatly appreciated. Done.*

3) line 87: the explicit definition of the term $IPG_\alpha$ should be introduced

*At lines 158-160 of the revised manuscript we have provided a small comment for the vertical discretization of the internal pressure gradient and the formulas which are the standard ones that can be found in [Shchepetkin and McWilliams, Journal of Geophysical Research, 2003 formula 2.3] or [Klingbeil et al., Ocean Modelling, 2018 formula 7.5]. More in general, we have provided the formulas for all the remaining terms appearing on the right-hand side.*

4) line 107: in formula (8), the argument of the flux limiter should be specified; this should also be done

in all the other points where this quantity is introduced, most of the time without specification of the argument and of the location at which it is computed; also in formula (9), a $\phi_\alpha$ appears that is not defined anywhere

*These formulas disappeared with the appendix.*

5) line 114: in formula (9), the second derivatives on the right hand side have no reason to be positive, so that this kind of upper bound should be performed considering only absolute values of the quantities involved

*Thank you for the correction, it is a mistake. We have removed the Appendix and formula (9) of the submitted manuscript with it.*

6) line 135-139: what the authors denote as the space discrete and fully discrete variables, respectively, are indeed (see e.g. formula (17)) the $P^1$ finite element approximation of the solution, which is a piecewise polynomial continuous function; the text should be changed to avoid this confusion; even though the notation $u_h(x)$ is customary in the finite element literature, it is a bit confusing in a context where h has a different meaning (the finite element 'h' would correspond to what in the preprint is called $\Delta x_E$.)

*We think the confusion comes from the fact that we wanted to describe the $z-$ surface-adaptive coordinate regardless of the horizontal spatial discretization (finite volume, finite element). And then some definitions conflicted with the finite element notation introduced for the tracer constancy verification. In the revised manuscript we have discussed only the case of first order staggered finite element implemented in SHYFEM. We have simply omitted $(\cdot)_h$ for the discrete quantities.*

7) line 146: the first sentence of section 3.1 is superfluous, any time discretization method will update the free surface based on equation (3)...the sentence should either be removed or reformulated if something else was meant

*We have removed such a trivial statement.*

8) line 158: coefficients $l_{\alpha,i}$ are introduced without having been previously defined; this is related to point 1) in the specific comments above; clear definition of these quantities is essential, since otherwise the proposed methods are not completely defined nor reproducible

*See answer to Specific comment 3)*

9) line 188: it is unclear what do the authors mean by 'z-layer depth at rest $\Delta z_\alpha^0$ ; if this is the depth at the initial time, it is better to say so because what 'at rest' means in a hydrodynamical simulation is very unclear

*With "at rest" we meant the reference configuration (see above, specific comment 3) which for us is always the one with $\zeta(x) = 0$ ("at rest"). Actually, it's true that the notation with $\Delta z_\alpha^0$ was not clear because it seems to suggest the initial time. Since in the revised manuscript we have introduced a reference domain with capital letter Z and its vertical discretization with layers of thickness $\Delta Z_\alpha$, we will simply replace the ambiguous $\Delta z_\alpha^0$ or $z_{\alpha-1/2}^0$ with the well defined quantities $\Delta Z_\alpha$, $Z_{\alpha-1/2}$. A picture has been added to help the reader.*

10) line 217: the expression 'hanging interfaces' is probably derived from the 'hanging nodes' used in the literature on numerical methods for non conformal meshes; however, while hanging nodes makes sense (the quadrature nodes on one side do not have a counterpart on the other side and numerical fluxes or mortar procedures must be employed), hanging interfaces does not make much sense in my opinion, since the interface is a perfectly well defined geometrical object; the authors are strongly suggested to modify this terminology and use instead e.g. non-conformal boxes, as they do in the following

*We agree that, in the literature, "hanging interface" never appears. We have removed this expression from the revised manuscript. In order to identify the geometrical objects we have used the expression "hanging*

*point" and "hanging layer". In line 440-442 of the revised manuscript: "Borrowing the vocabulary from the literature on non conformal meshes, we have a vertical edge with an hanging point. We call an hanging layer a layer for which at least one interface ends with an hanging point."*

13) line 244: the formula below this line is obtained according to the authors by integration by parts, but contains no boundary terms, the authors should explain whether these terms are zero and why or correct the formula

*At line 159 of the revised manuscript we have added a comment on the boundary integral. The contribution of this integral cancels out with the one of the neighbours element. It is not zero only at the lateral domain boundary but we have preferred to not discuss the implementation of boundary conditions.*

14) line 286: the rule used to define the mesh size should be explicitly reported, e.g. $h_K$ as the maximum length of the triangle sides

*When introducing the grid notation, at line 220 of the revised manuscript, we have defined more preicisely the element mesh size to which we refer in the test cases.*

16) in the caption of Figure 8, the quantities $T$, $T_0$ used to define the relative tracer conservation error are not defined; if they are meant to be the total tracer mass at the end and at the beginning of the simulation, then $|T - T_0|$ (absolute value is missing in the text!) is the absolute error, not the relative error; the authors are suggested to display values of $|T - T_0|/|T_0|$

*The caption of figure (8) was quite chaotic. The not defined quantity $T - T_0$ was meant to be the local tracer error. The absolute value was missing in the caption because it was then applied in the table with $||T - T_0||_{L1} = \int \int |T - T_0| \, dxdz$. Also, although the error shown was relative, it was not normalized in the caption. In the revised manuscript we have defined properly the relative error at line 475. The caption has been modified. We have rerun the test case with the last version getting exactly the same results shown in the table. Based on a remark from Reviewer 2, we have changed the mass/volume-conservation error. We have preferred to use the "barotropic" error (the volume error of the water column) instead of the layerwise error (the volume error of the fluid box). To avoid confusion the relative mass-error has been defined at line 473. In the revised manuscript the mass-error is less noisy probably because it is normalized by the volume of the water column and not of a single fluid box.*

17) line 437: replace 'its' with 'her' !!! it would also be appropriate to specify better the direct or indirect contribution of Dr. Bellafiore to this work

*Done. We are really sorry, this is a bad mistake due to english deficiency.*

**Reply to Reviewer 2**

**General comments**

1) The paper is currently suffering from a lack of mathematical rigor and the presentation is sometimes confused. The notation is often unnecessarily complex and some definitions are missing. Please refer to the comments in the attached PDF document and revise the presentation (especially in Section 2).

*In the revised version, we have restructured Section 2. First we have presented the multilayer (or layerwise) shallow water model for stratified fluids [Audusse et al.: Approximation of the hydrostatic Navier-Stokes system for density stratified flows by a multilayer model: Kinetic interpretation and numerical solution, J. Comput. Phys., 230, 3453–3478, 2011] with improved references, notation and details. There are some small differences with respect to this reference but they are clearly mentioned in the text. The position of the interfaces $z_{\alpha\pm1/2}$ and the consequent definition of the layers thickness $h_\alpha$, necessary to close the problem, is properly given. Then, as suggested by reviewer 1, we have given a clear definition of $h_\alpha$ also for $z-$star layers. Everything now is treated in a discrete vertical framework.*

*Concerning your main remarks related to Section 2 in the pdf:*

1.1) Line 60, unfortunately I found this section a little confused and had trouble following the presentation. The mathematical rigor of this section should be improved. Also, please add a few more references when you feel you are short on space, to provide more details.

*See above*

1.2) Line 65, could you add a figure to clarify the schematics?

*Thank you for the suggestion. In figure 1 of the revised manuscript we have added an illustration with the multilayer notation, similar to the one in [E. Audusse et al. A Multilayer Saint-Venant system with mass exchanges for shallow water flows. Derivation and numerical validation. ESAIM, 2011], but with a reversed layer ordering. We have added the same illustration also for $z-$star in figure 2 of the revised manuscript.*

1.3) Line 65, if alpha starts from 1, should this be z_3/2?

*Yes $\alpha$ starts from one but $\zeta = z_{1-1/2}$. Instead $z_{1+1/2} = z_{2-1/2}$ denote the first interior interface. You can check Figure 2 of the revised manuscript.*

1.4) Line 66, does this mean the origin of your reference frame is at the free surface?

*You can check figure 2 and 3 of the revised manuscript where the reference axis $x - z$ are sketched with dashed lines.*

1.5) Line 68, I do not understand why the x-dependence is explicit only for $b$

*We made explicit the $x-$dependence whenever help to understand.*

1.6) Line 70,71,72. what does this mean? who is z star?

*This part has been rewritten in a vertical discrete framework. All these continuous definitions have been removed. Moreover we have focused only on $z-$star. The discrete equivalent are now at lines 189 and 191 of the revised manuscript.*

1.7) Line 75. how is S defined for z-coord?

*For z-layer the interface position is given at line 210 of the revised manuscript (we do not need to define S in this case).*

1.8) Line 82, are these definitions or identities?

*They are definitions. We have just renamed the velocity of the interface $\partial_t z_{\alpha-1/2}$ that comes from the derivation of [E. Audusse et al. A Multilayer Saint-Venant system with mass exchanges for shallow water flows. Derivation and numerical validation. ESAIM, 2011] as $\sigma_{\alpha-1/2}$.*

1.9) Line 86, this notation is a bit odd, please let alpha start from 1

*Done*

1.10) Line 87, what is the meaning of this notation? I assume it is the same notation used to evaluate integrals but in any case please define

*Thank you for this remark. The square brackets notation means the difference between the value at upper and lower interface of a layer. This was not defined in the submitted manuscript. In the revised one we have defined the functions at the interfaces as well as their difference at lines 102-107.*

1.11) Line 87, this notation is a little awkward, maybe use a single symbol rather than an acronym?

*Done, we have used $B_\alpha$.*

1.12) Line 93, please find another term, momenta is not used usually

*In the revised manuscript we changed "momentum/transport" to "discharge". Momentum is used only referred to the "momentum equation".*

1.13) Line 97, please do not use a lowercase t for tracers, it is visually too similar to time

*In the revised manuscript we have used the capital letter $T_\alpha$ as in (Audusse,2011)*

1.14) "the so-called tracer constancy", this word is probably not what you want to use

*The name tracer constancy preservation (line 98 and 227) has been taken from [Shchepetkin and McWilliams, Ocean Modelling, 2005]. However, reviewer 1 pointed out that in the literature this is known as "compatibility with the continuity equation". We have modified the revised manuscript given the references suggested by reviewer 1.*

1.15) Line 101, "space" instead of "location"

*Done, Line 209 of the revised manuscript.*

2) The authors are doing a good job in the numerical results with showing that the proposed method is well behaved and that it does not perform worse than z-star. Although it is my opinion that additional results are missing to show that the extra hassle of introducing or removing layers is actually beneficial for some application. The authors should include at least one additional test case to show that their method enables simulations that are currently not possible with other existing z-type coordinate models. If this is deemed unfeasible, please show at least that there exists one relevant test case for which your model clearly outperforms existing z-coordinate models.

*You have raised a good point. We agree with you that this part was missing in the submitted manuscript. We would have expected some performance differences between $z-$surface-adaptive and $z-$star coordinate, in computations of intertidal flats. This is because the vertical discretization of the water column differs significantly in the two cases (with $z-$surface-adaptive one mainly performs runup/rundown with a one-layer shallow water model, with z-star the initial number of layers remains constant during the simulation). For the test #2 (tidal channel) such performance differences are still not very clear to us.*

*The advantages of our method against $z-$star are the classical advantages of using $z-$layers. The truncation error related to the pressure gradient term is very weak. Perhaps, having the cross-layers velocity that coincides with the vertical velocity (see line 214 of the revised manuscript) may be useful in regions with low frequency internal motion [Leclair and Madec: $\widetilde{z}$-coordinate, an Arbitrary Lagrangian–Eulerian coordinate separating high and low frequency motions, Ocean Modelling, 37, 139–152, 2011]. In fact z-star*

*layers are moving to follow the free-surface in an arbitrary manner with respect to this internal flow. This may cause larger truncation errors. We have commented these aspects in the revised manuscript. In light of the limited differences observed in the Venice Lagoon test case, we have searched for a new test case. We have substituted it with the SHYFEM application to the Po Delta. More significant disparities between the two methods can be observed in reproducing salinity stratification and salt water intrusion. Altough we did not have the time to validate these results against observations, we see that the z-surface-adaptive layers provides sharper results in terms of salt water intrusion and river plume. The validation is left for future work.*

3) The computational performance of the method is relegated to the last 4 lines of the numerical results section. Parallel performance is clearly of paramount interest for your readers so a more detailed analysis should be included in the paper. Currently, in the serial runs reported, the overhead of the present method is 8% over z-star. Would that get worse in parallel? If so, what are possible avenues for mitigation?

*We have not covered parallel implementation aspects because all the tests have been accomplished with a serial run. Although a recent version of SHYFEM is parallelized with MPI, the branch with the z−surface-adaptive developments only supports a partial parallelization with OMP. Unfortunately, this means that we have not been able to evaluate parallel performances of the z−surface-adaptive algorithm in such a short term. We have commented that the algorithm (grid movement, insertion/removal) mainly operates on the vertical grid, and the parallel execution of these tasks should not encounter any issue. The stencil of the numerical scheme is not enlarged with respect to the standard method. However some variables have been introduced only for the insertion/removal operations. This is the case of the nodal top layer index (Eq. 34 of the revised manuscript) which must be exchanged between the domains.*

**Remaining remarks in the pdf**

1) Line 6, "reverted", maybe "reduced to"?

*Done*

2) Line 13, "with the interfaces that follow the materials". I am not sure this is the right word

*Changed to "with the interfaces that are material surfaces".*

3) Line 48, move this right after you mentioned the first drawback at the top of the page, otherwise the reader can't follow

*The sentence no longer appears in the revised manuscript because the corresponding analysis, as suggested by reviewer 1, has been removed. We have slightly modified the introduction. First we speak of the advantages of z−layers, then the drawbacks, then what we do to overcome some of these drawbacks.*

4) Line 56, "... Shallow Water ..." I don't think you need to capitalize these

*Done*

5) The picture is not very informative because z-star and quasi-z look the same. Can you display a case where this does not happen?

*We have restricted the attention to the two vertical coordinates used in the numerical experiments: z−layers and z-star. Thus, the picture has been removed.*

6) Line 129, this itemized list makes the presentation a bit weird

*We have removed the list and reformulated this part, see lines 336-337 of the revised manuscript.*

7) Line 133, I believe you mean the horizontal domain. The domain is only 1D in the horizontal right? Otherwise it's 2D (1 horizontal and 1 vertical)

*In Section 2 the governing equations have been presented in a vertical discrete form without any explicit*

*dependence on the coordinate z. Here we proceed with the horizontal and time discretization. To avoid confusion we have denoted the fluid domain $\Omega$ and $\Omega_{\boldsymbol{x}}$ its projection on the horizontal plane, see Line 219 of the revised manuscript. Please note that we have restructured also this part. To avoid any ambiguous symbol, in the revised manuscript, we have focused only on the SHYFEM discretization (staggered Finite Element on triangular grids), while in the submitted manuscript the discussion was in 1D and general (Finite Element, Finite Volume).*

8) Line 135, you did not ever introduce the concept of a dual cell

*We think that the geometrical object called dual cell is introduced and defined: "The dual cell $C_i$ is obtained by joining the barycenters of the triangles in $D_i$ with the midpoints of the edges meeting in $i$". In the revised manuscript we have added a figure and the definition of dual cell area is in an equation and not in the text. We hope this clarifies this point.*

9) Line 136, did you define this?

*Borrowing the symbol from the Finite Element literature, this was meant to be the discrete approximation of a continuous variable. We apologize for the confusion of this part. As answered to Reviewer 1 the problem came from the fact that we wanted to keep the discussion general. In the revised manuscript we treat only the semi-implicit finite element case in Section 3 where we provide clear definitions and statements about the horizontal and temporal discretization of the multilayer model.*

10) Line 141, does alpha start from 0 or 1?

*From one.*

11) Line 141, what do you mean by z-grid?

*We meant a vertical grid composed with z-layers. We recognize that it is unusual and in the revised manuscript we have removed it.*

12) Line 143, why not just use $h_\alpha^0$?

*Yes, that could be an option. However we have preferred to use the delta notation. We mention that, after a remark of Reviewer 1, we have changed the notation for the reference grid ("at rest"), in the revised manuscript. Instead of the superscript $\cdot^0$, which can generate confusion with the initial time, we use capital letters Z to distinguish the reference configuration from the physical one. This is explained in Section 2.*

13) Line 150, what do yo mean by "spanned" here?

*In the revised manuscript we have clarified this part: "we identify the indices associated to the layers that, locally, undergo a deformation. They are defined as the layers of the reference grid whose top-interface finds above the free-surface or by the set of indices: ..."*

14) Line 153. "$\epsilon_{mov}$ can be used to control the number of moving layers". How exactly?

*Thank you for this remark. This was only explained after in Section 3.3 of the submitted manuscript. We have removed this sentence as, at this point of the discussion, it only serves to create further confusion. How to use this parameter to move all the layers is still explained in Section 4.3 of the revised manuscript.*

15) Line 154, would it be possible to simplify this notation?

*Done, $\alpha_{movBot}$ is $N_{mov}$ in the revised manuscript.*

16) Line 158, I know you mean "implies" with this arrow so maybe just write it.

*We have removed these formulas that were just weighing down the text. More in general, in the revised manuscript we have removed all the arrows that mean "imply".*

17) Line 168, this is obtained subtracting appropriate z values as defined in (11) right? if so please make

that explicit

*Yes exactly. This definition was not present in the submitted manuscript. In Section 2 of the revised manuscript we have given precise definitions on how to compute the layer thickness, see Line 91 or 193 of the revised manuscript.*

18) Line 160, this notation is not exactly rigorous...

*We wanted to point out that the multilayer equations are time-stepped on the vertical grid with the surface interface that are locally moving. This ambiguous sentence has been removed in the revised manuscript.*

19) Line 165, please use indices

*Done*

20) Line 175, I am not following this. In a pseudo time? Line 178, how is this defined?

*This part need a clarification. Our approach consists in considering the area swept by the interface in one time-step as the sum of two contributions: one due to the grid movement with velocity $\sigma_{mov}$ and one due to the collapse of the element with grid velocity $\sigma_{top}$, see fig. 4 of the revised manuscript (in the submitted manuscript unfortunately both the interface velocities have the identical symbol $\sigma$, this has been changed). The pseudotime is $\tau = (t - t^n)$. It has been introduced because, instead of solving the governing equations (9 and 10 of the revised manuscript) at once with $\sigma = \sigma_{mov} + \sigma_{top}$, we have considered a splitting procedure. First we solve the governing equations on a moving grid, that is the discrete counterpart of eq. (9) or (10) with $\sigma_{mov}$. Then we solve equation (38) (always of the revised manuscript) with $\sigma_{top}$. We have reviewed this part explaining carefully these concepts. The pseudo time was not necessary and it has been removed.*

21) Line 187, "the parameter $\epsilon_{mov}$ that prescribes the number of moving surface layers." actually, above you said "fixes the minimum allowable depth for a layer"

*We agree that this sentence is misleading. In the revised manuscript we have clarified this part. Basically we meant that, if $\epsilon_{mov}$ is small it is used to avoid very thin layers in the grid movement step. Playing with $\epsilon_{mov}$, it can be also used to move all the layers, not only the surface ones.*

22) Line 191, $r_\alpha$ how does this relate to $r_{mov}$?

*$R_\alpha$ (we switched to capital letter in the revised manuscript) is a real value that can be computed for each $\alpha = 1, N$. If we set $r_{mov} = R_N$ then it is easy to check from Eq.(35) in line 374 and line 377 of the revised manuscript that $N_{mov,i} = N$. At all nodes $i$ all layer will be moving. We used this trick to perform the z-star computations.*

23) $r_{top}$ what is this new parameter????

*It has been defined before; in the revised manuscript it appears at line 365. It is used to remove the surface fluid boxes before the layer thickness at some node of the triangle reaches a very small value. In fact, if we put $r_{top} = 0$, we have experienced problems with the diffusion matrix which has $h_\alpha$ at the denominator.*

24) Line 205, "it is the old one, the new one?" please remove, too informal

*Done*

25) Line 223, $K^*$ the meaning of this is not yet introduced.

*Done*

26) "of preserving the constancy property" I am not sure this is the word here. Maybe "invariance"?

*See reply at point 1.14*

27) Line 235, maybe use discharge instead of momenta?

*Done*

28) Line 235 and 240 "form functions", "kernel". Do you mean basis functions?

*Yes we do, in the revised manuscript we have used the word "basis"*

29) Line 238, I think you mean the characteristic function on E

*Yes we do. We have changed it.*

30) Line 238, Line 290 "currents", velocities? please use velocity, if that is what you mean

*In the revised manuscript we have used the word "velocity" everywhere in the text.*

31) Figure 6, why are these cells colored?

*At the bottom, $z-$layers models apply a mask to non-existing land boxes that make the bathymetry stepped These colored cells are such masked boxes. In the revised manuscript, in the caption, we have added "stepped bathymetry with masked boxes in brown"*

32) Line 252 and Line 345, "always", you mean again?

*Thank you. In the revised manuscript we have changed such an improper use of the word "always".*

33) Figure 7, why is the free surface crossing layers below?

*In this figure we have reported both the free-surface and the reference $z-$layers. It was to highlight that when the free surface crosses the first layer below it means that layer, locally, has been removed from the computation. So, for the finer resolution (red colour) at $t = 0.3\,s$ six layers has been removed in the central part of the domain. In the revised manuscript we have improved the caption.*

34) Figure 8, what is causing this bump in the error in your opinion for the case of 24 layers?

*Thank you for observing this behaviour. We are considering relative mass/volume errors: the volume of fluid lost/gained by a fluid box is divided by the fluid box volume. This can be very small for surface layers that are prone to be removed. We believe this may cause these bumps. In the revised manuscript we have changed the error definition and we have used the "barotropic" relative error (the volume error of the water column normalized by the volume of the water column). To avoid confusion the relative volume-error has been defined at line 473 of the revised manuscript. The barotropic volume error is less noisy because, we believe, it is normalized by the volume of the water column.*

35) Line 308, the slope should not depend on x and is negative in your case (-10/25).

*You are correct! The revised manuscript has been modified.*

36) Line 360. In the numerical results, you are showing that the method you propose works and that it is not worse than z-star. But is it better? What is the motivation for using it? I may have misunderstood but it looks like there is no clear indication in the paper that the method you propose has tangible superior qualities over other strategies. Can you produce a test case where a certain simulation can only be possible with the insertion and removal of layers? Right now, I think you are presenting a very interesting and well crafted algorithm, but without enough evidence that the extra hassle of implementing layer addition/removal is actually justified.

*See reply to point 2.*

---

## Author Response (AR2)

**Reply to Reviewer 1**

1) on page 3, line 80, the coefficients s are denoted as functions, which is misleading since it is then stated just below that these are constants. The word 'functions' should be replaced with 'coefficients'.

*Done*

2) Even though the paper's results do not depend on this, for completeness, on page 7, around line 160 an equation of state should be introduced, since no relationship between density and temperature is given anywhere else.

*We have elucidated the relationship between density and salinity using the UNESCO equation of state at line 98.*

3) on page 7, line 159, a nabla operator is apparently missing in the definition of the b gradient at intermediate levels

*Thank you. We have corrected the typo.*

4) the statement 'The $z-$layers are a particular case where the interfaces do not depend on time and space' on line 206 of page 9 is unclear. From the previous definitions of the Z terms, these do not seem to be dependent on time, so it is unclear why the z-layer case should be a particular time independent case. This point should be clarified.

*We refer to the actual interfaces, not to the reference one. Contrary to the models introduced earlier, the actual interfaces $z_{\alpha \pm 1/2}$ do not depend on time. In the text, we have been more precise about this point, without alluding to "particular cases".*

5) Starting on line 268 of page 11, the authors should introduce some changes in the discussion of the semi-implicit time discretisation. Firstly, stating as done by the authors that semi-implicit time discretizations are standard for ocean models is not correct. Several important ocean models use split explicit time discretizations. In my opinion, semi-implicit methods are superior for a number of easily provable mathematical reasons, but still this is not really the standard in this literature. Furthermore, the advantage of using two different implicitness parameters is debatable. The use of two different thetas makes the presentation more complicated, while the parameter values seem to have always been taken equal to each other. I would suggest to use a single theta. In order to help the reader following the derivation, it would also be appropriate to introduce in this context the fully discrete mass equation, which is instead presented only later. Finally, the authors should state that the time discretisation they use is a well known one, referring explicitly to the widely cited paper "Casulli, Vincenzo, and E. Cattani. "Stability, accuracy and efficiency of a semi-implicit method for three-dimensional shallow water flow." Computers & Mathematics with Applications 27.4 (1994): 99-112." in which it has essentially been first introduced in the community of coastal and ocean modelling.

*Following your suggestions we have made small changes. We have removed the expressions "as it is standard for ocean models" and "popular choice for many coastal ocean models". Since the introduction, we have added the reference about the semi-implicit method for the shallow water flows together with the reference for the staggered finite element, on which SHYFEM relies. Moreover, the text has been rewritten with one single $\theta$.*

6) On line 364 of page 15, the authors state that the problem with thin vertical layers is that the vertical diffusion matrix becomes ill-conditioned. This seems to be the least important issue, compared to the stability restrictions of the explicit advection discretisation. Removing thin layer is important for a number of reasons, but the authors are suggested to provide different motivations for doing so, also because really extreme ratios between the thickest and thinnest layer should be reached for this conditioning issue to be a serious one.

*Thank you for the remark and for elucidating this point. We have tried to run some of the tests without*

*vertical viscosity (in order to exclude the ill-conditioning issue). Without the removal operations, we had to reduce strongly the time step otherwise oscillations in the velocity appear near thin layers. In figure we share the results for the impulsive wave test. We have changed the discussion about this point.*

Figure title: horizontal velocity profiles at $t = 0.5\,s$

with layer removal, $\Delta t = 0.005$     no layer removal, $\Delta t = 0.005$     no layer removal, $\Delta t = 0.0001$

7) On line 522 of page 23 the statement ' Flooding is thus performed with a 1-layer shallow water model with the classical wetting/drying algorithms that may be deployed in dry or nearly dry areas (e.g. positivity limitation, discharge regularization, etc...).' is unclear and should be clarified with respect to the previous and the following statements. Do the authors refer only to the 1d model that is used as a reference? From the way the statement is formulated, it seems that this 1d approach is also used in the z-adaptive method, but it is unclear how, since as discussed before this statement the z-adaptive method inserts more layers as the simulation goes on. Before more layers are introduced (e.g. in the wetting phase) one could imagine that the same 1d approach is used, but it is unclear how this can be done in the drying phase, when multiple layers are present.

*As you said the sentence is referred only to the flooding phase. For the drying phase, under a CFL condition based on the vertical velocity, we believe that also drying should occur with one layer only. However, due to presence of finite thresholds for the identification of the dry elements, it may happen that, for very high vertical resolutions, more layers become dry in one time step. We have thus removed the ambiguous sentence. Moreover, the dry region in SHYFEM is treated in a simplified manner: a mass-conserving barotropic momentum is computed from a "flattened" free-surface and it is then distributed across the layers. Within this simplified algorithm, having more layers does not present any specific issue.*

8) The results of the realistic simulation of the Po delta are interesting and promising. However, it is a but awkward to refer to a mesh with 24 layers as 'coarse', as opposed to a 'fine' mesh with 27 layers. While it is clear that 27 layers are enough to trigger the adaptation algorithm, it would be appropriate to remove these 'coarse' and 'fine' labels and, possibly, to perform further simulations with an even larger number of layers, thus using a mesh that can honestly be called 'fine' with respect to the 24 layer one.

*We have removed the "coarse" and "fine" labels. We were not able to embark on a complicated convergence study.*

9) On line 566, a possible larger truncation error of the vertical advection scheme is mentioned. It is unclear to what the 'larger' refers, since all the advection schemes employed are first order upwind.

*Yes, the order of accuracy is the same for both the runs. However the truncation error depends also on the constant that multiplies the grid spacing, which in turn depends on the advection velocity. The mass-transfer function may have different values between the two runs: with the $z-$layer it coincides with the vertical velocity and with $z-$star it depends also the derivatives of the grid interfaces. We are speculating that (it's merely a conjecture) for the z-star run, the mass-transfer function is larger then the vertical velocity, triggering a larger error. We have tried to enhance the clarity of this possible explanation.*

Suggested minor corrections

*We have implemented all these minor corrections. We have verified the English using an online tool. We have corrected another typo in formula (18).*

**Reply to Reviewer 2**

The authors have massively revised the article and I am satisfied with the changes now. Thank you for addressing my concerns. I only have one minor question:

In figure 1 and 2, if alpha is one, then it looks like there are some surfaces with the same name. For instance in Figure 1, the free surface is $s_{1/2}$. If below you let alpha be one, then third surface from the top is also $s_{1/2}$. Can you address this?

*Thank you for the comment. We have added vertical ellipsis to stress the fact that $\alpha$ indicates a generic layer index between 1 (first layer) and N (last layer). The remaining notation in both figures seems correct to us. If the first layer is $\alpha = 1$, then the third surface from the top is $s_{3-1/2} = s_{5/2}$ and not $s_{1/2}$.*

Also, there is a typo on line 30, "the" should be "they".

*Done*